# Structural insights into the architecture and membrane interactions of the conserved COMMD proteins

Michael D Healy[1], Manuela K Hospenthal[2†], Ryan J Hall[1], Mintu Chandra[1], Molly Chilton[3], Vikas Tillu[1], Kai-En Chen[1], Dion J Celligoi[2], Fiona J McDonald[4], Peter J Cullen[3], J Shaun Lott[2], Brett M Collins[1]*, Rajesh Ghai[1]*

[1]Institute for Molecular Bioscience, The University of Queensland, St. Lucia, Australia; [2]School of Biological Sciences, The University of Auckland, Auckland, New Zealand; [3]School of Biochemistry, Biomedical Sciences Building, University of Bristol, Bristol, United Kingdom; [4]Department of Physiology, University of Otago, Dunedin, New Zealand

*For correspondence:
b.collins@imb.uq.edu.au (BMC);
r.ghai@uq.edu.au (RG)

Present address: †Institute of Structural and Molecular Biology, Department of Biological Sciences, Birkbeck College, London, United Kingdom

Competing interests: The authors declare that no competing interests exist.

**Abstract** The COMMD proteins are a conserved family of proteins with central roles in intracellular membrane trafficking and transcription. They form oligomeric complexes with each other and act as components of a larger assembly called the CCC complex, which is localized to endosomal compartments and mediates the transport of several transmembrane cargos. How these complexes are formed however is completely unknown. Here, we have systematically characterised the interactions between human COMMD proteins, and determined structures of COMMD proteins using X-ray crystallography and X-ray scattering to provide insights into the underlying mechanisms of homo- and heteromeric assembly. All COMMD proteins possess an α-helical N-terminal domain, and a highly conserved C-terminal domain that forms a tightly interlocked dimeric structure responsible for COMMD-COMMD interactions. The COMM domains also bind directly to components of CCC and mediate non-specific membrane association. Overall these studies show that COMMD proteins function as obligatory dimers with conserved domain architectures.
DOI: https://doi.org/10.7554/eLife.35898.001

## Introduction

The COMMD (Copper Metabolism Murr1 (Mouse U2af1-rs1 region 1) Domain) proteins are highly conserved in metazoans and unicellular protozoa (*Burstein et al., 2005*; *Maine and Burstein, 2007*). There are ten family members that play key roles in intracellular trafficking and in the regulation of transcription (*Burstein et al., 2005*; *Maine and Burstein, 2007*). A hallmark of the COMMD family members is a highly conserved C-terminal sequence of ~70–80 amino acids called the COMM domain, which has no known structure. The N-terminal domain of these proteins, which we refer to as the HN (helical N-terminal) domain, is more variable in sequence across the ten proteins, and is proposed to ascribe unique functions to the different family members (*Burstein et al., 2005*; *Maine and Burstein, 2007*). Despite the high degree of conservation and the important roles of COMMD proteins in membrane trafficking and cell signalling, little is known about their structures or their specific molecular functions.

In humans, all ten members of the COMMD family (Commd1-Commd10) are expressed broadly in many tissues (*Burstein et al., 2005*; *van De Sluis et al., 2002*). Much of our current understanding of the COMMD proteins is derived from studies of the founding family member Commd1. An in-frame deletion in exon 2 of the *COMMD1* gene was first identified in Bedlington terrier dogs with

hepatic copper toxicosis (*van De Sluis et al., 2002*; *Tao et al., 2003*). In accordance with the link of Commd1 to copper metabolism, it was reported to bind to Cu(II), implying that COMMD family members might serve as metal binding proteins (*Narindrasorasak et al., 2007*), although this was questioned by subsequent studies. Commd1 has also been shown to associate with both ATP7A and ATP7B and is thought to mediate their trafficking from endosomes to the plasma membrane (*Materia et al., 2012*). In humans, ATP7A and ATP7B are responsible for controlling intracellular Cu (II) accumulation (*Tao et al., 2003*; *Materia et al., 2012*; *Vonk et al., 2011*; *Miyayama et al., 2010*; *Phillips-Krawczak et al., 2015*) and are involved in conditions related to copper toxicity including Wilson's disease (*Wang et al., 2011*), suggesting an alternative role for Commd1 in copper homeostasis. In line with a more general function in intracellular trafficking, Commd1 depletion causes a marked reduction in surface levels of low-density lipoprotein receptor (LDLR), with increased endosomal accumulation leading to a concomitant increase in cholesterol levels (*Bartuzi et al., 2016*). Although COMMD proteins are expected to be structurally similar, there may be non-redundant trafficking roles mediated by different members of the family. For example, Commd9 is deleted in patients with Wilms tumour-aniridia syndrome (WAGR) syndrome, and appears to interact specifically with Notch2 to regulate its surface abundance by shuttling the receptor away from the lysosomal degradative pathway (*Li et al., 2015*). Members of the COMMD family have also been linked to nuclear factor kappa-light-chain-enhancer of activated B cells (NF-κB) transcriptional regulation, hypoxia adaptation, and regulation of intracellular sodium concentration via interactions with the epithelial sodium channels (ENaCs) (*de Bie et al., 2006*; *Liu et al., 2013*; *Maine et al., 2009*).

COMMD proteins have been found to exist in an endosome-associated complex with the coiled-coil domain-containing proteins CCDC22 and CCDC93, which has been termed the CCC (COMMD, CCDC22, CCDC93) complex. In addition, functional proteomics and bioinformatics have independently shown that the CCC complex can associate with a stable heterotrimeric assembly of VPS29, C16orf62 and DSCR3, to form a larger macromolecular complex dubbed the 'Commander' or CCC/Retriever complex (*Mallam and Marcotte, 2017*; *Wan et al., 2015*; *McNally et al., 2017*). Other proteins such as SNX17, RanBP1, SH3GLB1 and FAM45a are also thought to associate with the Commander complex (*Wan et al., 2015*; *Dey et al., 2015*; *Hein et al., 2015*; *Huttlin et al., 2015*). Further complexity arises from evidence that COMMD proteins can form homo- and heterodimers through their COMM domains (*Burstein et al., 2005*). Although Commd9 and Commd5 associate directly with each other (*Wan et al., 2015*), it remains unclear whether the COMMD proteins interact physically with each other more generally, or how this relates to their assembly with CCDC22 and CCDC93 in CCC and the larger Commander complex.

In this study we provide a comprehensive and systematic analysis of the structural, biophysical and biochemical properties of the COMMD family of proteins, and their recruitment to the CCC complex. These analyses show that COMMD proteins form family-wide homo- and heterodimers through their C-terminal COMM domains. The crystal structure of the COMM domain from Commd9 reveals an intimately interlocked α/β homodimeric structure, which involves formation of a large and conserved hydrophobic interface. The crystal structure of the N-terminal domain of Commd9 shows an all α-helical structure, and using small angle X-ray scattering we establish that homodimers of full-length Commd1, Commd7 and Commd9 adopt similar tertiary architectures. Intriguingly, the COMMD proteins show clear structural homology to a poorly characterised protein found in chlamydial species of bacteria. Further biochemical studies show that C-terminal COMM domains have pleiotropic roles in both recruitment of CCDC22 and CCDC93 via their N-terminal calponin homology-like (NN-CH) domains, and in binding non-specifically to negatively charged phospholipids.

## Results

### COMMD proteins form homodimers via the C-terminal COMM domain

As a first step towards assessing the self-assembly activities of COMMD proteins, we studied the biophysical properties of recombinant full-length COMMD proteins in solution. Of all COMMD proteins we tested Commd1, Commd7 and Commd9 (*Figure 1—figure supplement 1*) were readily purified and tractable for biophysical analyses. We measured their molecular weights by analytical size exclusion chromatography coupled with multi-angle laser light scattering (SEC-MALLS). This showed that Commd1, Commd7 and Commd9 form homodimers and are monodisperse in solution

(*Figure 1A,B*). SEC-MALLS data suggests that dimerization is likely to be a general property of the COMMD family members. To determine the domains required for homodimerisation we performed SEC-MALLS analyses of isolated HN and COMM domains. While the Commd1 and Commd9 HN domains behave as monomers, Commd1 and Commd9 COMM domains exist as dimeric species (*Figure 1B,C*). This data shows that the COMM domain is both required and sufficient to mediate COMMD protein dimerization.

## Structural analysis of the COMMD proteins

We next sought to determine the X-ray crystallographic structures of COMMD proteins. However, although crystals of various full-length COMMD proteins grew rapidly, their diffraction quality was

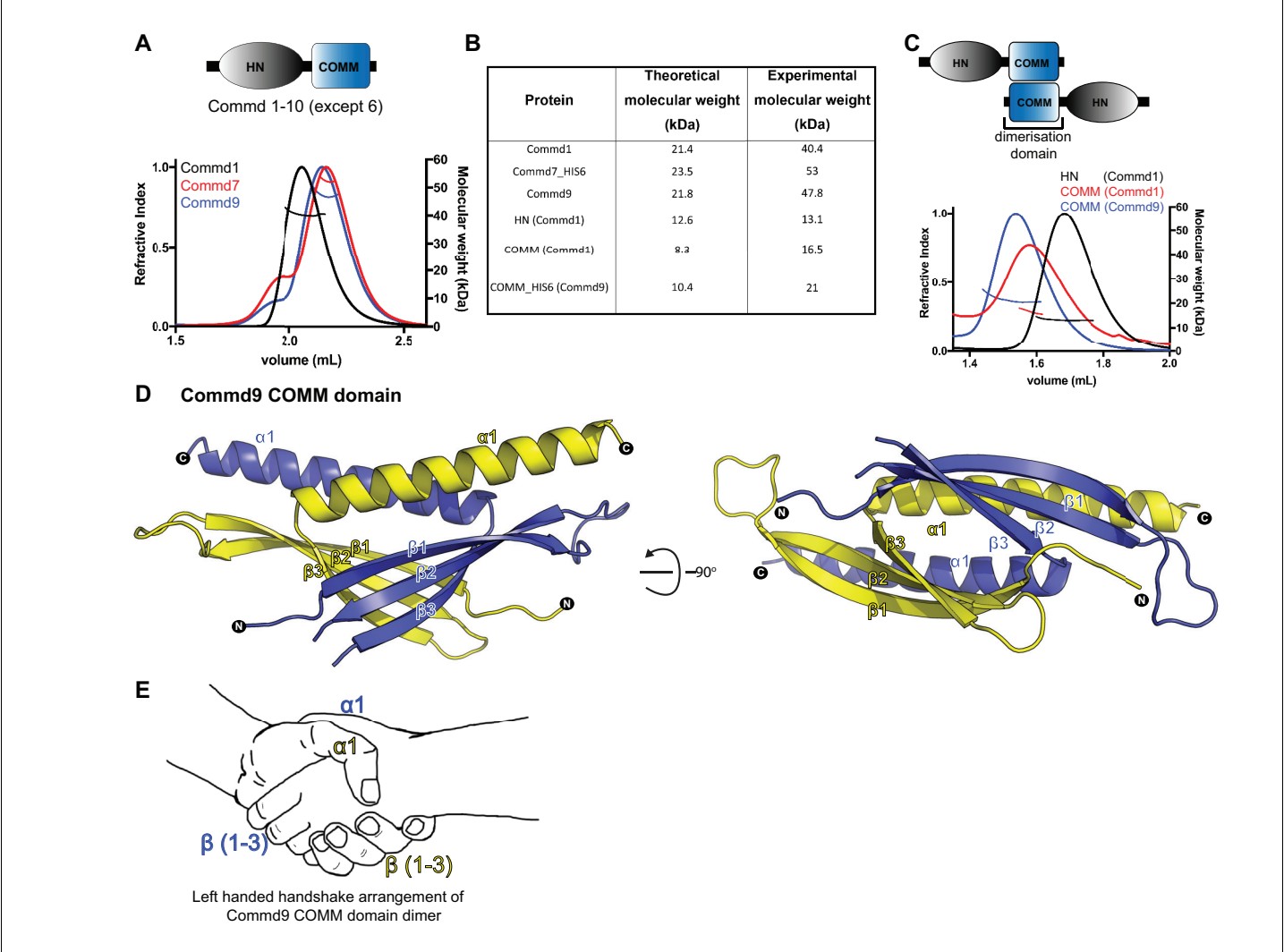

**Figure 1.** COMMD proteins dimerize through the C-terminal COMM domain. (A) Cartoon representation of COMMD proteins, and the MALLS profile of Commd1 (black), Commd7 (red) and Commd9 (blue) showing Commd proteins are dimers in solution (B) Comparison of the theoretical molecular weight of the COMMD proteins and the experimentally measured molecular weight. (C) MALLS analyses of the COMM and HN domain shows that the protein dimerisation occurs through the C-terminal COMM domain of the COMMD proteins as represented schematically. (D) Ribbon representation of the dimeric Commd9 COMM domain (residues 115–198) chain A (yellow) and chain B (blue). (E) The structure of the COMM domain dimer is analogous in orientation to a left-handed handshake.

DOI: https://doi.org/10.7554/eLife.35898.002

The following figure supplement is available for figure 1:

**Figure supplement 1.** Schematic representations of bacterial expression constructs used in this study.

DOI: https://doi.org/10.7554/eLife.35898.003

not sufficient for structure determination. Therefore a divide-and-conquer approach was taken to determine structures of the two individual domains of the protein, the conserved C-terminal COMM domain, and the variable N-terminal HN domain.

### Crystal structure of the Commd9 C-terminal COMM domain

We determined the crystal structure of the Commd9 COMM domain to 2.2 Å resolution by single-wavelength anomalous dispersion (SAD). The overall structure of the Commd9 COMM domain is composed of two cone-shaped chains that are tightly intertwined with each other to form a globular dimeric module (*Figure 1D*, *Table 1*). Each monomer is comprised of an N-terminal three-stranded β-sheet capped by an α-helix, with the overall arrangement making an open, hairpin-like structure. A simple analogy for the COMM domain dimer is that it resembles a left-handed handshake, where the sheet and helix from each monomer represent the interlocked palms and thumbs of each hand respectively (*Figure 1E*). The overlapping C-terminal α-helices of each chain bury a large hydrophobic surface area of approximately 2100 Å$^2$ (nearly 1/3$^{rd}$ of the monomer surface area) supporting the notion that the native state of COMMD proteins is to form dimers via the COMM domain.

**Table 1.** Summary of crystallographic structure determination statistics.*

|  | Commd9 HN domain (1–117) |  | Commd9 COMM domain (115–198) |
| --- | --- | --- | --- |
| Data collection |  |  |  |
| Space group | P 1 |  | I 4 |
|  | 28.5 Å, 35.6 Å, 54.3 Å; |  | 79.4 Å, 79.4 Å, 58.5 Å |
| Unit cell dimensions (a,b,c; α,β,γ) | 104.0°, 93.3°, 91.1° |  | 90°, 90°, 90° |
|  | Remote | Inflection |  |
| Wavelength (Å) | 0.9762 | 0.9796 | 0.9787 |
| Total reflections | 172817 | 146841 | 131332 |
| Resolution range (Å) | 19–84 – 1.55 (1.58) | 19–84 – 1.55 (1.58) | 47.13–2.17 (2.24) |
| Mean I/sigma(I) | 16.9 (5.2) | 25.2 (9.3) | 26.7 (4.9) |
| R-merge | 0.10 (0.54) | 0.06 (0.18) | 0.06 (0.5) |
| Unique reflections | 27698 | 27442 | 9689 |
| Multiplicity | 6.2 (6.0) | 5.4 (5.3) | 13.6 (11.8) |
| Anomalous Multiplicity | 3.1 (3.1) | 2.6 (2.7) | 6.9 (6.1) |
| Mn(I) half-set correlation CC(1/2) | 0.99 (0.89) | 0.99 (0.98) | 0.99 (0.95) |
| Completeness (%) | 91.9 (54.9) | 91.0 (52.7) | 99.7 (96.4) |
| Anomalous Completeness (%) | 89.6 (50.3) | 88.1 (48.1) | 99.3 (93.0) |
| Wilson B-factor | 8.4 | 18.6 | 53.1 |
| Refinement |  |  |  |
| R-work |  | 0.13 (0.25) | 0.24 (0.30) |
| R-free |  | 0.16 (0.33) | 0.28 (0.35) |
| Resolution range (Å) |  | 26.28–1.55 | 30.36–2.17 |
| Number of atoms |  | 2078 | 1268 |
| Protein atoms |  | 1784 | 1259 |
| RMS(bonds) |  | 0.009 | 0.002 |
| RMS(angles) |  | 1.158 | 0.549 |
| Ramachandran favored (%) |  | 100 | 98.06 |
| Ramachandran outliers (%) |  | 0 | 0 |
| Average B-factor |  | 13.11 | 85.61 |

*Highest resolution shell is shown in parentheses.

DOI: https://doi.org/10.7554/eLife.35898.004

## COMMD proteins dimerise via a conserved hydrophobic interface in the COMM domain

In *Figure 2* the interface that mediates Commd9 dimerisation is examined in closer detail. Using CONSURF (*Ashkenazy et al., 2016*; *Landau et al., 2005*), the most evolutionarily conserved residues in Commd9 were mapped onto the structure. There is a very high degree of sequence conservation seen in the hydrophobic core of the Commd9 COMM domain, particularly along the α-helical surface mediating dimerisation (*Figure 2A*). Multiple sequence alignment of the COMM domains of all the COMMD proteins demonstrates conservation of many hydrophobic amino acids, particularly leucine residues (*Figure 2B*). The high degree of conservation, and high degree of hydrophobicity within the dimerization interface strongly suggests that all of the COMMD proteins will form obligatory dimeric structures through similar mechanisms. We attempted to mutate several interfacial

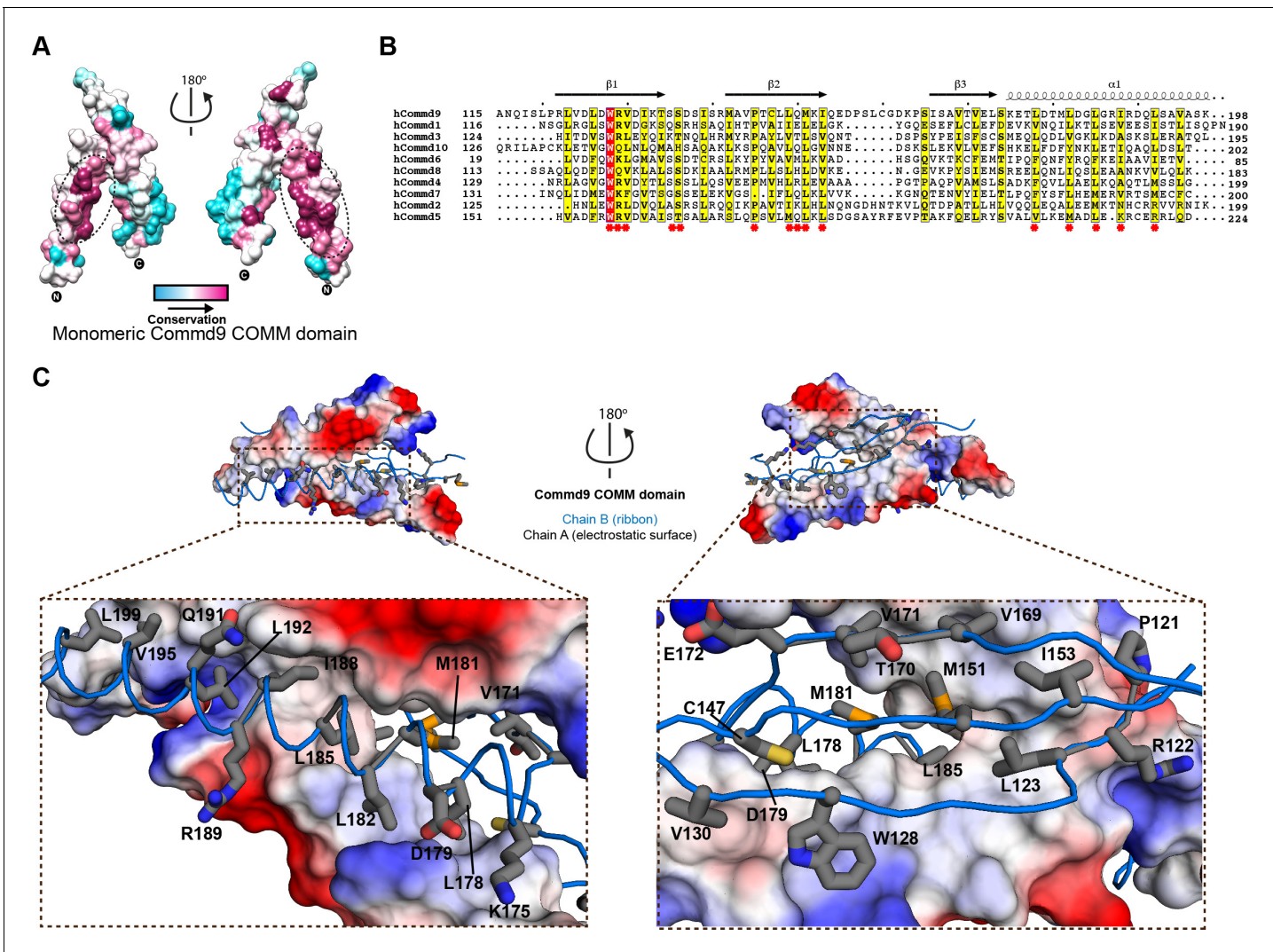

**Figure 2.** Structural mechanism underpinning the COMMD dimerization. (A) A surface map of the conserved and variable residues of the Commd9 COMM domain showing the hydrophobic core is highly conserved while the surface residues are more variable, confirming the importance of dimerization for COMMD stability. These calculations are made using the Consurf server (*Ashkenazy et al., 2010*) on chain A. (B) A combined sequence alignment and secondary structure comparison of COMM domains of all the human COMMD proteins highlights that the C-terminal COMM domain is highly conserved across the COMMD family of proteins. Residues marked by asterisk depict conservation of amino acids that decorate the dimerization interface. Alignments were made with ESPript 2.2 (http://espript.ibcp.fr/ESPript/ESPript/) (*Gouet et al., 2003*) (C) Representation of dimer interface of Commd9 COMM domain highlighting key residues (chain A, worm in blue) making main chain-main chain, stacking, salt bridge interactions with the electrostatic surface (chain B).

DOI: https://doi.org/10.7554/eLife.35898.005

residues, but both the full-length mutated COMMD proteins and isolated COMM domains were aggregated due to poor protein solubility (not shown). This provides further support for the essential nature of dimer formation.

## Crystal structure of the Commd9 N-terminal HN domain

We next determined the structure of the Commd9 N-terminal domain by X-ray crystallography to a resolution of 1.55 Å using multi-wavelength anomalous dispersion. Overall, the Commd9 N-terminal domain has a globular architecture and is composed of a six-helix bundle with a meander topology (*Figure 3A*, *Table 1*). We therefore refer to this domain as the HN (*H*elical *N*-terminal) domain. The overall structure of the HN domain has a similar all α-helical fold to the equivalent domain of Commd1, which was previously determined by NMR (*Figure 3B*) (*Sommerhalter et al., 2007*). Compared to Commd1 however, the HN domain of Commd9 has an additional α-helix at its N-terminus that packs down on top of the structure, and a large disordered loop in Commd1 is better resolved in Commd9. In Commd9 the α4 helix forms a central core element that extends the length of the HN domain. In Commd1 however, the equivalent helix is bent and oriented differently. This could be

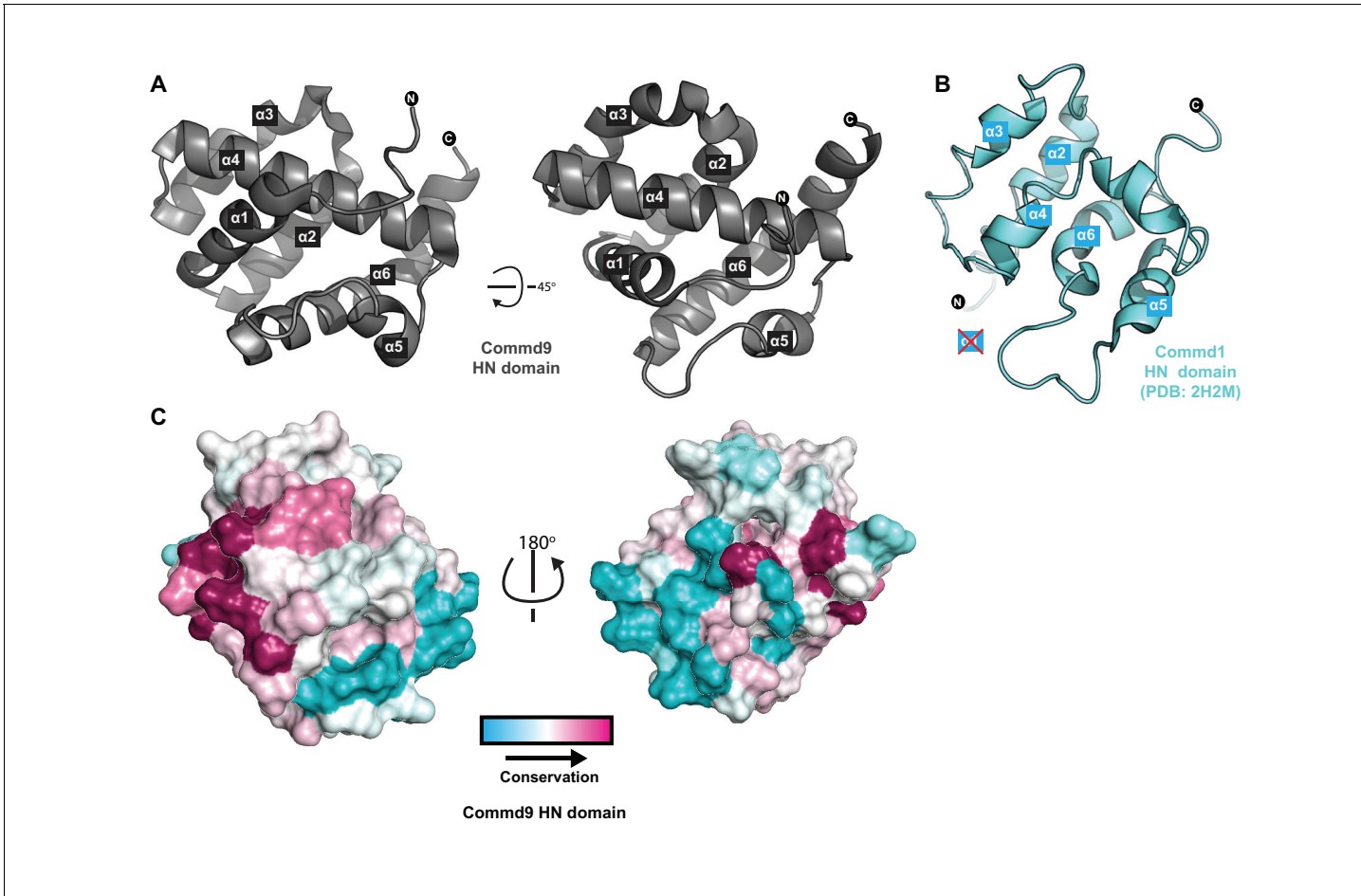

**Figure 3.** Structure of the N-terminal HN domain of Commd9. (A) Cartoon representation of Commd9 HN domain (1–116) crystal structure crystallized at pH 4.9 showing a globular structure. (B) NMR structure of the Commd1 HN domain (PDB ID 2H2M) (*Sommerhalter et al., 2007*) shown in cartoon diagram in the same orientation as Commd9 in *Figure 3A* left panel. The Commd1 HN domain lacks the first helix (α1) and appears to have a kinked helix α4, but overall shares the same topology. (C) Sequence conservation of Commd9 HN domain mapped on the structure using CONSURF.
DOI: https://doi.org/10.7554/eLife.35898.006

The following figure supplement is available for figure 3:

**Figure supplement 1.** Sequence alignments of HN domain of all the human and zebrafish COMMD proteins.
DOI: https://doi.org/10.7554/eLife.35898.007

due to the absence of the α1 helix to provide stability, or might also be due to an insufficient number of restraints in this region used for NMR structure calculations. However, the topology of the five shared α-helices (α2-α6) is similar overall. Comparing the HN domains of Commd1 and Commd9 using DALI (*Holm and Laakso, 2016*; *Holm and Rosenström, 2010*) showed structural similarity (DALI Z-score >3) with an RMSD of 3.2 Å over 83 Cα atoms.

The electrostatic surface of the Commd9 HN domain reveals the presence of two basic patches and a negatively charged region, but it is not clear yet what their functional significance might be (*Figure 3—figure supplement 1A*). Mapping sequence conservation of the Commd9 HN domain across species shows conserved surface residues mainly in the α1 helix region (*Figure 3C*). The N-terminal regions of the COMMD proteins are quite variable in sequence across the ten family members (although they are well conserved in paralogous proteins across species), and this has led to it being referred to as a variable domain or VARD (*Maine and Burstein, 2007*). Bioinformatics and secondary structure analyses of the human COMMD proteins however, shows that in all of the family members the HN domain is very likely to share the same α-helical topology, as well as across various species of Commd9 (*Figure 3—figure supplement 1B,C*). In support of this, the HN domain of human Commd3 shows a strong α–helical l signal when examined using far-UV circular dichroism spectroscopy (not shown). The exception to this is human Commd6, which does not possess an HN domain at all, although Commd6 orthologs in other species such as fish and amphibians do (*Figure 3—figure supplement 1C*).

## Solution structures of COMMD proteins reveal a conserved homodimeric structure

In the absence of high-resolution crystal structures of full-length COMMD proteins, we employed SEC-SAXS to obtain solution structural information regarding the architectures of homodimeric Commd1, Commd7 and Commd9 proteins (*Figure 4*; *Figure 4—figure supplement 1*; *Figure 4—figure supplement 2*; *Table 2*). The scattering curves and pair distribution functions (*P(r)*s) both signify a relatively globular structure. *Ab initio* structures of these calculated from the SAXS data reveal compact but elongated molecules. We next performed rigid body fitting of the SAXS data with the program SASREF (*Petoukhov and Svergun, 2005*), using the crystal structures of the Commd9 HN and COMM domains for Commd7 and Commd9 whereas Commd1 HN and Commd9 COMMD domain was used for generating the Commd1 model. The resulting models obtained by this approach were superimposed on the SAXS envelope. The theoretical scattering profiles of Commd1, Commd7 and Commd9 models, and the *ab initio* molecular envelopes are in good agreement with the experimental scattering data with low $\chi^2$ values. These studies suggest that the COMMD proteins, including Commd9 are homodimers with an elongated shape in solution.

## COMMD proteins are structurally related to a unique protein from chlamydial species

The COMMD proteins share no detectable sequence homology to any other proteins. However, structural comparison of the Commd9 COMM domain to the structures in the Protein Data Bank (PDB) using DALI (*Holm and Laakso, 2016*; *Holm and Rosenström, 2010*) did identify low scoring (DALI Z-score >2.5) structural matches to the PH domain of human pleckstrin as well as the phox homology (PX) domain (DALI Z-score >3) of yeast Grd19. Interestingly, it is clear from these comparisons that the COMM domain has a similar topology to core fragments of the larger PX and PH structures (*Figure 4—figure supplement 3*). A closer structural match however was identified with the Pur-α (purine-rich element binding protein) repeat domain (DALI Z-score >4.5), a whirly-like nucleic acid binding fold (*Graebsch et al., 2009*; *Weber et al., 2016*) (*Figure 4—figure supplement 4A*). Overlay of the Commd9 COMM domain with repeats of Pur-α reveals a similar fold with a RMSD of 2.5 Å over nearly 50 Cα atoms.

Intriguingly, the clearest structural matches to Commd9 are two closely related proteins from the bacterial species *Chlamydia trachomatis* (CT584) and *Chlamydia pneumoniae* (Cpn0803) (*Barta et al., 2013*; *Stone et al., 2012*) (DALI Z-scores > 7.5). CT584 and Cpn0803 are orthologous proteins found only in chlamydia. They are modular proteins with an α-helical N-terminal domain and an α/β C-terminal domain, both of which are structurally analogous to the respective HN and COMM domains of Commd9 (*Figure 4—figure supplement 4B,C*). The only major difference in the

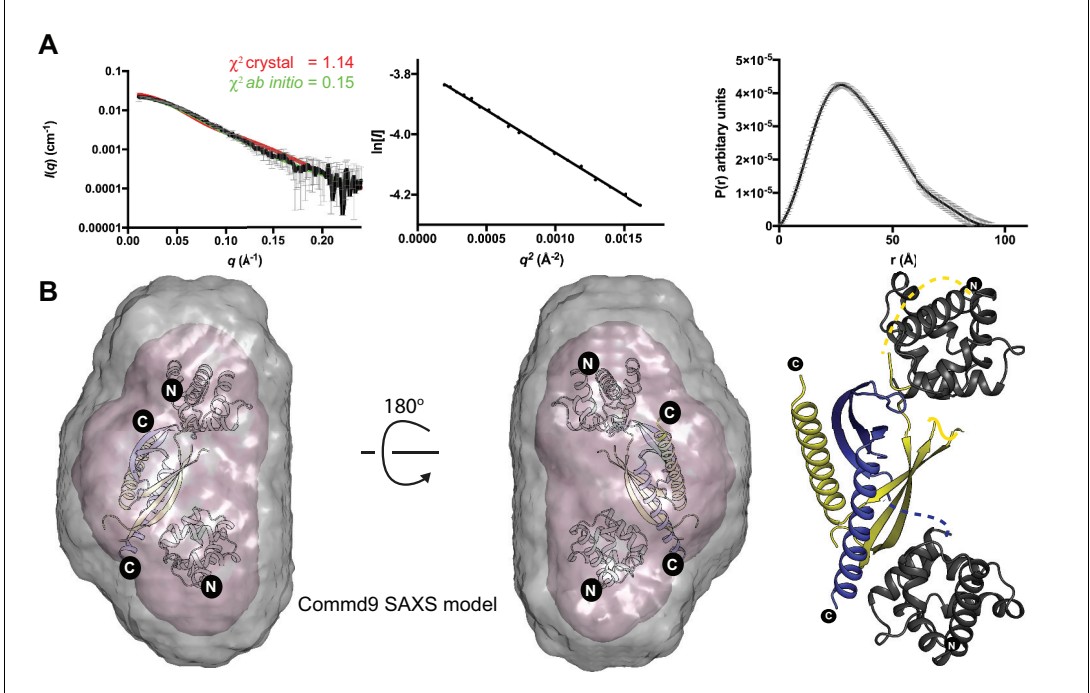

**Figure 4.** The solution structure of Commd9 determined by SAXS. (**A**) (Left) Experimental scattering profile Commd9 in black overlaid with the theoretical scattering curve calculated from the Commd9 rigid body model (red) and the *ab initio* model determined by GASBOR (green) using CRYSOL. (Middle) The Guinier plot for the experimental data at the low-angle region ($q_{max} \times R_g < 1.3$). (Right) $P_{(r)}$ functions derived from the SAXS data. (**B**) (Left) Averaged (grey) and filtered (coral) molecular envelopes from GASBOR. The *ab initio* model of Commd9 was docked with the rigid body model of Commd9 using SUPCOMB. (Right) Model of Commd9 protein from the experimental SAXS shown in ribbon diagram with HN domains in grey and the COMM domain dimer in blue and yellow.

DOI: https://doi.org/10.7554/eLife.35898.008

The following figure supplements are available for figure 4:

**Figure supplement 1.** SEC-SAXS profiles of Commd1, Commd7 and Commd9.
DOI: https://doi.org/10.7554/eLife.35898.009

**Figure supplement 2.** The solution structures of Commd1 and Commd7 determined by SAXS.
DOI: https://doi.org/10.7554/eLife.35898.010

**Figure supplement 3.** Structural comparison of the COMM domain of Commd9 with other similar modules.
DOI: https://doi.org/10.7554/eLife.35898.011

**Figure supplement 4.** Structural similarity of COMMD proteins to Pur-α repeats and chlamydial proteins.
DOI: https://doi.org/10.7554/eLife.35898.012

C-terminal structures is the presence of two additional α-helices at the C-terminus of the bacterial proteins (*Figure 4—figure supplement 4B*). The N-terminal domains of the chlamydial proteins also have an analogous overall topology to the HN domain, except that the bacterial proteins lack α1 and α5 (*Figure 4—figure supplement 4C*). Like the COMMD molecules the chlamydial proteins possess a core dimeric structure formed by the C-terminal domains, with an overall architecture that is very similar to that of Commd9 determined by SAXS (*Figure 4—figure supplement 4D*). In addition both chlamydial proteins appear to form hexameric assemblies via trimerisation of the core dimer structure (*Figure 4—figure supplement 4E*) (discussed further below).

## COMMD proteins bind promiscuously to each other

There are now a number of high-throughput proteomics studies that point to the existence of a large multi-subunit assembly containing all of the COMMD family proteins (*Mallam and Marcotte, 2017*; *Wan et al., 2015*; *McNally et al., 2017*; *Dey et al., 2015*; *Hein et al., 2015*; *Huttlin et al., 2015*). This is also well supported by several more targeted studies demonstrating that COMMD proteins associate with each other in both endogenous and over-expression conditions (*Wan et al.,*

**Table 2.** Summary of SAXS structural parameters.

| | Commd1 | Commd7 | Commd9 |
|---|---|---|---|
| **Data collection parameters** | | | |
| Instrument | Australian Synchrotron SAXS/WAXS beamline with Dectris PILATUS 1M detector | | |
| Wavelength (Å) | 1.0332 | | |
| Beam geometry (μM) | 250 × 130 | | |
| Camera length (m) | 1.6 | | |
| $q$-range (Å$^{-1}$) | 0.011–0.251 | | |
| Absolute scaling method | Comparision with scattering from 1 mm H2O | | |
| Normalization | To transmitted intensity by beam-stop counter | | |
| Method for monitoring radiation damage | Dose maintained below 210 Gy | | |
| Exposure per frame (s) | 1 | | |
| Sample temperature (K) | 283 | | |
| Sample configuration | SEC-SAXS with sheath-flow cell | | |
| Flow-rate (ml/min) | 0.25 | | |
| **Structural parameters** | | | |
| *Guinier analysis* | | | |
| $I(0)$ (cm$^{-1}$) | 0.04 ± 0.00 | 0.03 ± 0.00 | 0.02 ± 0.00 |
| $R_g$ (Å) | 29.26 ± 0.25 | 28.86 ± 0.60 | 27.61 ± 0.79 |
| $q_{min}$ (Å$^{-1}$) | 0.01 | 0.01 | 0.01 |
| $qR_g\ max$ (Å$^{-1}$) | 1.3 | 1.3 | 1.3 |
| *P(r) analysis* | | | |
| $I(0)$ (cm$^{-1}$) | 0.04 ± 0.0004 | 0.029 ± 0.0003 | 0.02 |
| $R_g$ (Å) | 30.37 ± 0.59 | 29.5 ± 0.53 | 28.26 ± 0.67 |
| $D_{max}$ (Å) | 101.47 | 99.47 | 96.99 |
| $q$ range (Å$^{-1}$) | 0.011–0.251 | 0.011–0.251 | 0.011–0.251 |
| Porod volume (Å$^{-3}$) | 71951 | 65393 | 67180 |
| Dry volume calculated from sequence (Å$^{-3}$) | 25848 | 27271 | 26622 |
| **Shape model-fitting results** | | | |
| *Gasbor (default paramters, 20 calculations* | | | |
| $q$ range (Å$^{-1}$) for fitting | 0.011–0.251 | 0.011–0.251 | 0.011–0.251 |
| Symmetry, anisotropy assumptions | $P2$, none | $P2$, none | $P2$, none |
| NSD (standard deviation) | 0.16 (0.004) | 1.04 (0.55) | |
| $\chi^2$ range, CORMAP P values | 0.15–0.20, 0.11 | 0.14–0.17, 0.47 | 0.14–0.15, 0.14 |
| *Atomistic modelling* | | | |
| Crystal structures | PDB entry 2H2M and 6BP6 | PDB entry 4OE9 and 6BP6 | PDB entry 4OE9 and 6BP6 |
| $q$ range for all modelling | 0.011–0.251 | 0.011–0.251 | 0.011–0.251 |
| *CRYSOL (with default parameters)* | | | |
| No constant subtraction | | | |
| $\chi^2$ | 0.31 | 0.72 | 1.14 |
| Predicted Rg (Å) | 30.85 | 30.54 | 31.88 |
| Vol (Å), Ra (Å), Dro (eÅ$-3$) | 53643, 1.40, 0.007 | 46010, 1.400, 0.075 | 57201, 1.400, 0.007 |
| *FoXS* | | | |
| $\chi^2$, CORMAP P values | 0.52, 0.99 | 0.78, 0.99 | 0.99, 0.93 |
| Predicted Rg (Å) | 30.82 | 29.18 | 31.84 |
| $c_1,c_2$ | 1.03, 0.29 | 1.05, 4.0 | 1.04, 1.20 |

*Table 2 continued on next page*

*Table 2 continued*

|  | Commd1 | Commd7 | Commd9 |
|---|---|---|---|
| Molecular mass determination |  |  |  |
| Estimated MW from Porod volume | 41387 Da | 38094 Da | 39099 Da |
| Calculated MW from MALLS | 40400 Da | 53000 Da | 47800 Da |
| Calculated MW from sequence | 21362 Da (42724 Da for dimer) | 22540 Da (45080 Da for dimer) | 21819 Da (43639 Da for dimer) |
| Software employed |  |  |  |
| Primary data reduction | Australian Synchrotron SAXS/WAXS data reduction package (Scatterbrain) |  |  |
| Data Processing | PRIMUS and GNOM |  |  |
| *Ab initio* modeling | GASBOR |  |  |
| Validation and averaging | DAMAVER |  |  |
| Computation of models | SASREF |  |  |
| Envelope representations | PyMOL |  |  |

DOI: https://doi.org/10.7554/eLife.35898.013

*2015*; *McNally et al., 2017*). In the majority of cases these interactions were found using co-immunoprecipitation strategies, although it has also been shown that heterodimeric complexes are formed upon bacterial co-expression of Commd1-Commd6, Commd1-Commd5 and Commd9-Commd5 (*Wan et al., 2015*). To study these pairwise interactions systematically we initially attempted a GST pull down assay with purified His-tagged Commd1 and all of the GST-tagged COMMD members. However, we did not observe any significant interactions using this approach (not shown). Next, we assessed whether co-translation would lead to heterodimeric interactions. GST-baits (all COMMDs) were co-expressed in *E. coli* with selected His-tagged preys (Commd1, 7, 9 and 10) followed by affinity purification using glutathione sepharose beads (*Figure 5A*, *Figure 5—figure supplement 1*). Western blotting was used to confirm interactions unambiguously as GST and His-tagged COMMD proteins are similar molecular weights. GST-Commd8 was not included in these experiments due to the tendency of this protein to degrade. This assay reveals that all of the COMMD proteins are able to co-assemble with each other in a highly promiscuous manner, while the lack of binding using separately purified proteins suggests that no exchange occurs between pre-formed homodimeric proteins. In general there is little specificity seen in the heteromeric complexes formed, although Commd10 binds most strongly to Commd2 and Commd5 and only weakly with Commd9, while Commd9 binds weakly to Commd6.

Previous reports have shown that the COMM domain alone may be sufficient to allow interactions between COMMD proteins (*Burstein et al., 2005*). To test this we next performed our co-expression GST-pull down assays with His-tagged COMM domains of Commd1 and Commd9 and the HN domain of Commd1 as prey. While no interaction was observed between any COMMD proteins and the N-terminal domain of Commd1, strong interactions were seen for C-terminal COMM domains of Commd1 and Commd9 mirroring the full-length proteins (*Figure 5A*). Overall, our data indicates that COMMD proteins form promiscuous homo- and heterodimeric complexes through their C-terminal COMM domains. The fact that COMMD-COMMD interactions could only be reconstituted after co-translation suggest that these complexes most likely involve the formation of dimeric structures analogous to that seen in the Commd9 COMM domain crystal structure.

## Reconstitution of stable heteromeric COMMD complexes

Our GST pulldown experiment demonstrates that Commd10 preferentially binds to Commd2 and Commd5. To assess the stoichiometry of heteromeric Commd complexes, we co-expressed GST-Commd5 with Commd10-His and purified the complex by sequential affinity purification using glutathione sepharose and TALON beads. The eluted complex was subjected to size exclusion chromatography (SEC) to obtain a 1:1 stoichiometric complex. The SEC fractions under the peak clearly show reconstitution of a 1:1 Commd5-Commd10 complex (*Figure 5B*).

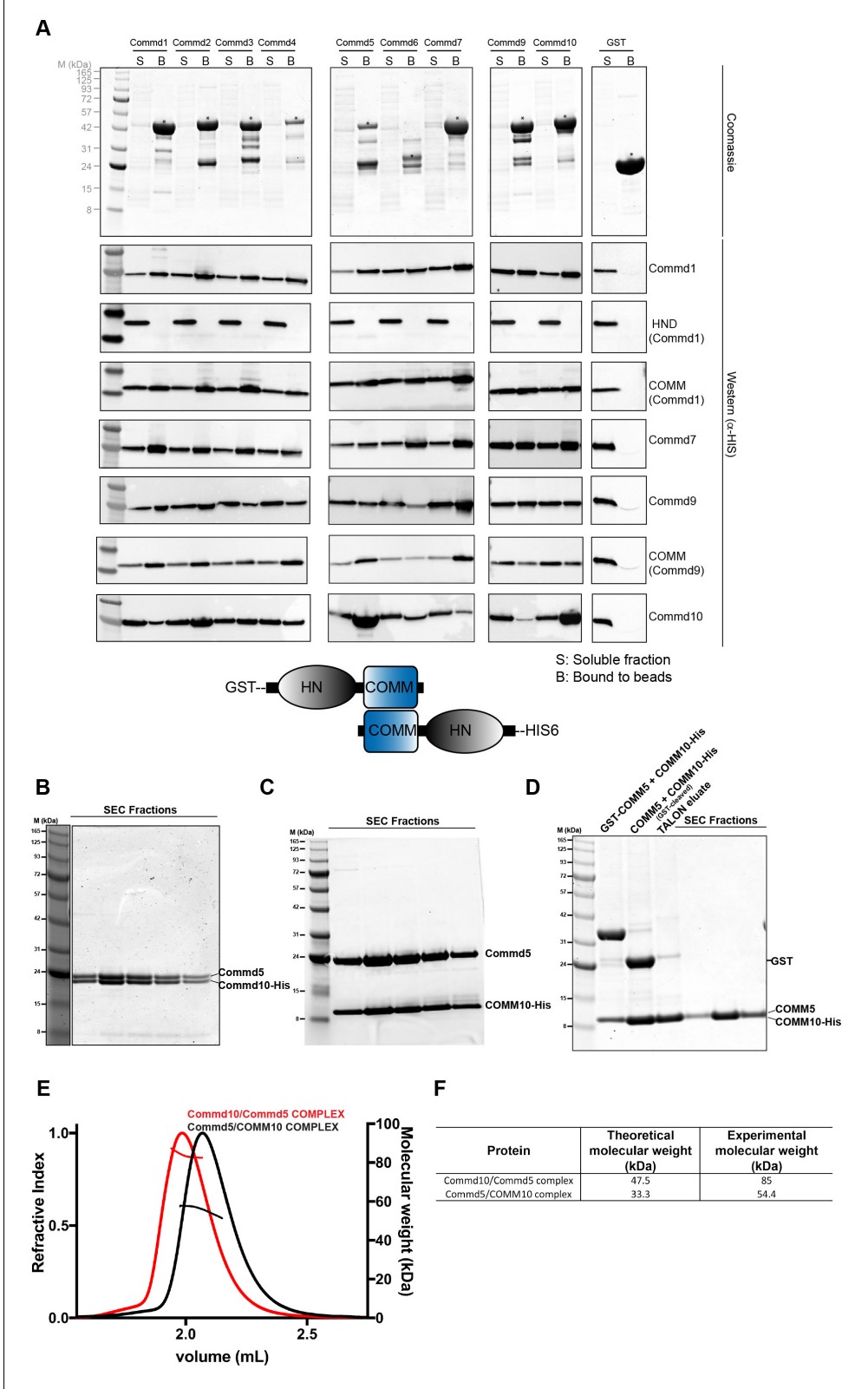

**Figure 5.** COMMD proteins form an array of homo and heterodimers via the C-terminal COMM domain. (**A**) A representative Coomassie image showing the relative expression of the GST fusion COMMD proteins (top). Blots probed with anti-His6 antibody demonstrating the direct pair-wise interactions of the COMMD protein family (bottom). A schematic representation of COMMD-COMMD interaction as demonstrated by the pull-downs. (**B, C, D**) Representative Coomassie images showing the reconstitution of full length Commd5-Commd10 complex, full length Commd5-Comm domain

*Figure 5 continued on next page*

*Figure 5 continued*

of Commd10 complex and complex of COMM domains of Commd5 and Commd10 respectively. (**E, F**) MALLS analyses of full length Commd5-Commd10 complex and Commd5-COMM domain of Commd10 complex showing the complexes potentially form heterotetramers based on the calculated molecular weight of the complexes.

DOI: https://doi.org/10.7554/eLife.35898.014

The following figure supplement is available for figure 5:

**Figure supplement 1.** Raw files of SDS-PAGE gels and western blots of COMMD-COMMD and COMMD-CCDC interactions.

DOI: https://doi.org/10.7554/eLife.35898.015

Since the COMM domain singularly mediates the formation of homo- and heteromeric complexes of COMMD proteins, we also attempted to make the complex using the full-length Commd5 and COMM domain of Commd10 and COMM domains of Commd5 and Commd10 respectively. Indeed, highly pure and stoichiometric species were isolated confirming the role of the COMM domain in COMMD complexes (*Figure 5C,D*). Several Commd proteins form COMM domain dependent homodimers and this prompted us to investigate the biophysical properties of Commd5-Commd10 complexes. SEC-MALLS analysis of Commd5-Commd10 reveals that the complex is highly monodisperse in solution. Although the combined theoretical molecular weight of these proteins is 47.5 kDa, the experimental mass was calculated to be 85 kDa, which suggests the existence of a Commd5-Commd10 heterotetramer (*Figure 5E,F*). Along the same lines, the Commd5-Commd10 COMM domain complex appears to also form a heterotetrameric assembly (*Figure 5E,F*). Altogether, these data raises two possibilities. First, COMM domains of Commd5 and Commd10 form homodimers and assemble together as a tetramer. Second, the Comm domain of Commd5 and Commd10 when expressed together form heterodimers ultimately forming the larger heterotetramer. In both of these scenarios, COMMD proteins are seemingly interacting with each other through a second binding interface on the COMM domain that is potentially distinct from the dimerization interface.

## The COMM domains bind the calponin homology domains of CCDC22 and CCDC93

Central to the endosomal trafficking function of COMMD proteins is their assembly into the CCC complex (*Mallam and Marcotte, 2017*; *Wan et al., 2015*). CCDC22 and CCDC93 are large proteins predicted to contain N-terminal divergent calponin homology domains (NN-CH) and C-terminal coiled-coils (*Schou et al., 2014*), and are critical components of the CCC complex (*Phillips-Krawczak et al., 2015*; *Bartuzi et al., 2016*; *Wan et al., 2015*). A Commd1 knock out causes loss of CCDC22/CCDC93 (*Phillips-Krawczak et al., 2015*; *Bartuzi et al., 2016*). Therefore, we set out to examine whether CCDC22/CCDC93 directly interact with COMMDs, and their mode of association. Using the co-translation pull down assay, we found that Commd1, 7 and 10 bind to both CCDC22 and CCDC93 directly (*Figure 6A*, *Figure 5—figure supplement 1*). Commd9 in contrast appears to recognize CCDC93 specifically. We next performed domain-truncations and conducted the binding assay with isolated calponin homology-like domains (NN-CH) and C-terminal coiled coil regions of CCDC22 and CCDC93. Interestingly, CCDC22 and CCDC93 bind to COMMDs chiefly through the NN-CH domain. Consistent with the previous literature (*Phillips-Krawczak et al., 2015*; *Starokadomskyy et al., 2013*), we also observed relatively weaker interaction bands for the C-terminal coiled coil domain of CCDC22 and CCDC93. Our data also shows that similar to COMMD-COMMD interaction, COMMD-CCDC22/CCDC93 binding occurs through the COMM domain.

In contrast to interactions between COMMD family members, which require co-expression for assembly, we find that CCDC22 and CCDC93 interaction with pre-formed COMMD dimers occurs spontaneously in vitro (*Figure 6B*). GST-pull down experiments conducted by mixing the purified GST-NN-CH domain of CCDC22 and CCDC93 with full length Commd1 and Commd9 as well as the COMM domains showed a similar binding pattern to what was observed after co-expression (*Figure 6A,B*). Specifically, Commd1 and Commd9 bind more strongly to the N-terminal NN-CH domain of CCDC93 in comparison to CCDC22. Moreover, this assay also suggests that Commd1 has stronger affinity than Commd9 for CCDC proteins. The interaction of the CCDC22 and CCDC93 NN-CH domains with the COMM domain of Commd1 was recapitulated and quantified using biolayer interferometry (BLiTz). A dose-dependent increase in the binding of the COMM

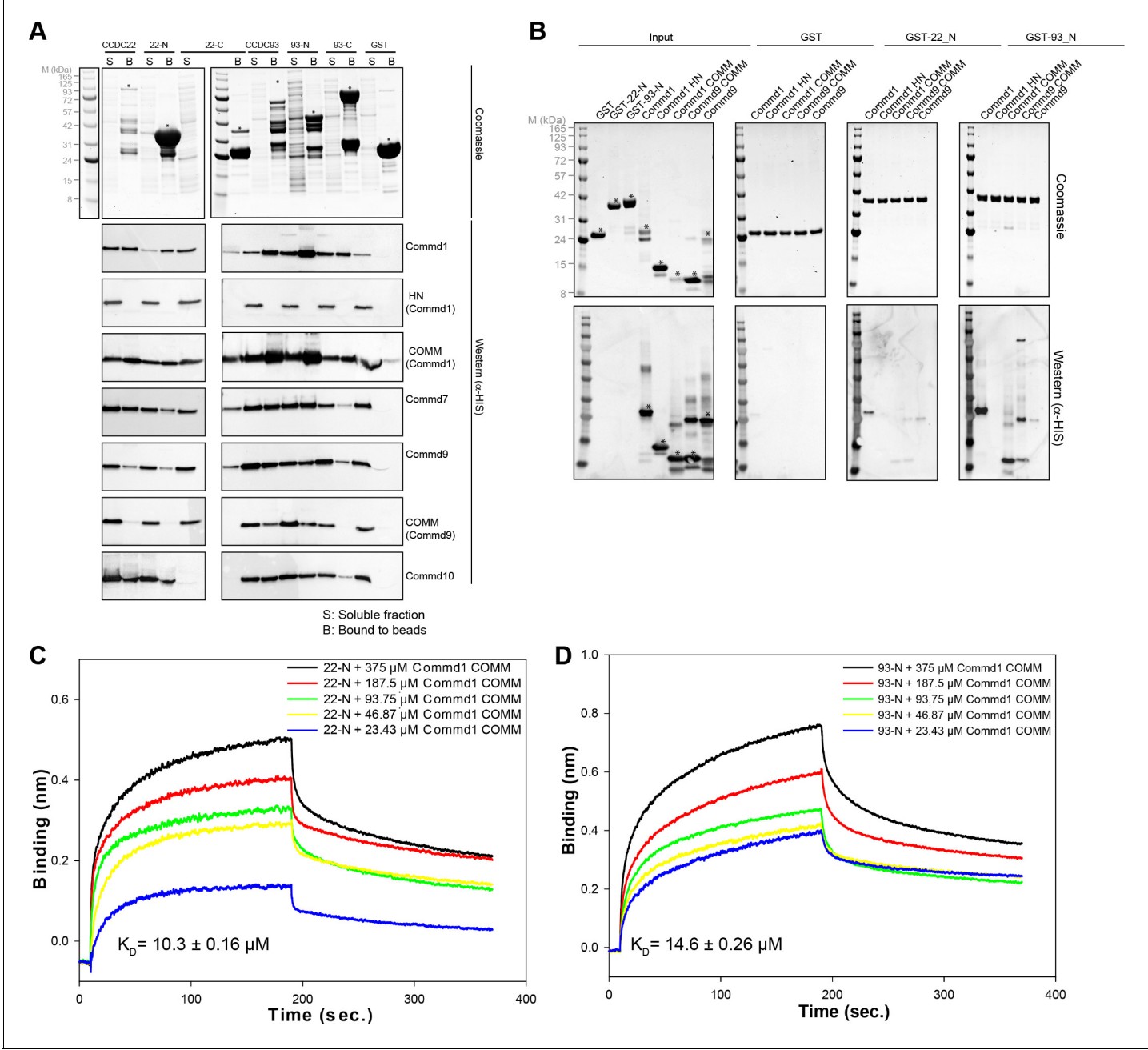

**Figure 6.** COMMD proteins bind directly to the N-terminal NN-CH domains of CCDC22 and CCDC93 via the COMM domain. (**A**) A representative coomassie gel of GST fusion protein expression in a co-expression set-up (Top). Anti-His6 blots of the GST pull down demonstrating direct interactions between the COMMD family proteins and the Coiled-coil domain proteins 22 and 93 (Bottom). (**B**) GST pull down demonstrating the binding capacity of the N-terminal domain of CCDC22 and 93 to Commd1 and Commd9 as well as their respective COMM domains. (**C** and **D**) Binding of the Commd1 COMM domain to the N-terminal domains of CCDC22 (**C**) and CCDC93 (**D**) measured at varying concentrations using the BLiTz system. Binding kinetics we calculated using Sigma Plot (Systat Software Inc.).

DOI: https://doi.org/10.7554/eLife.35898.016

domain of Commd1 with NN-CH of CCDC22 and CCDC93 was observed, and the affinity ($K_d$) was calculated to be 10.3 and 14.6 μM respectively (*Figure 6C,D*).

## Commd proteins bind to CCDC22 and CCDC93 via a conserved site

To identify the molecular determinants that govern the interaction between the COMM domain of COMMDs and NN-CH domain of CCDC22 and CCD93, we used cross-linking mass spectrometry (MS) in combination with pull down experiments. Non-deuterated BS3 cross-linker was used to crosslink full-length Commd9 with the NN-CH domain of CCDC93. Due to the availability of HN and COMM domain structures, Commd9 was chosen for these experiments. The NN-CH domain of CCDC22 could not be used as it does not contain any lysine residues. Upon crosslinking, three major bands (labeled as 1, 2 and 3) in the Commd9 and NN-CH domain of CCDC93 mixture were

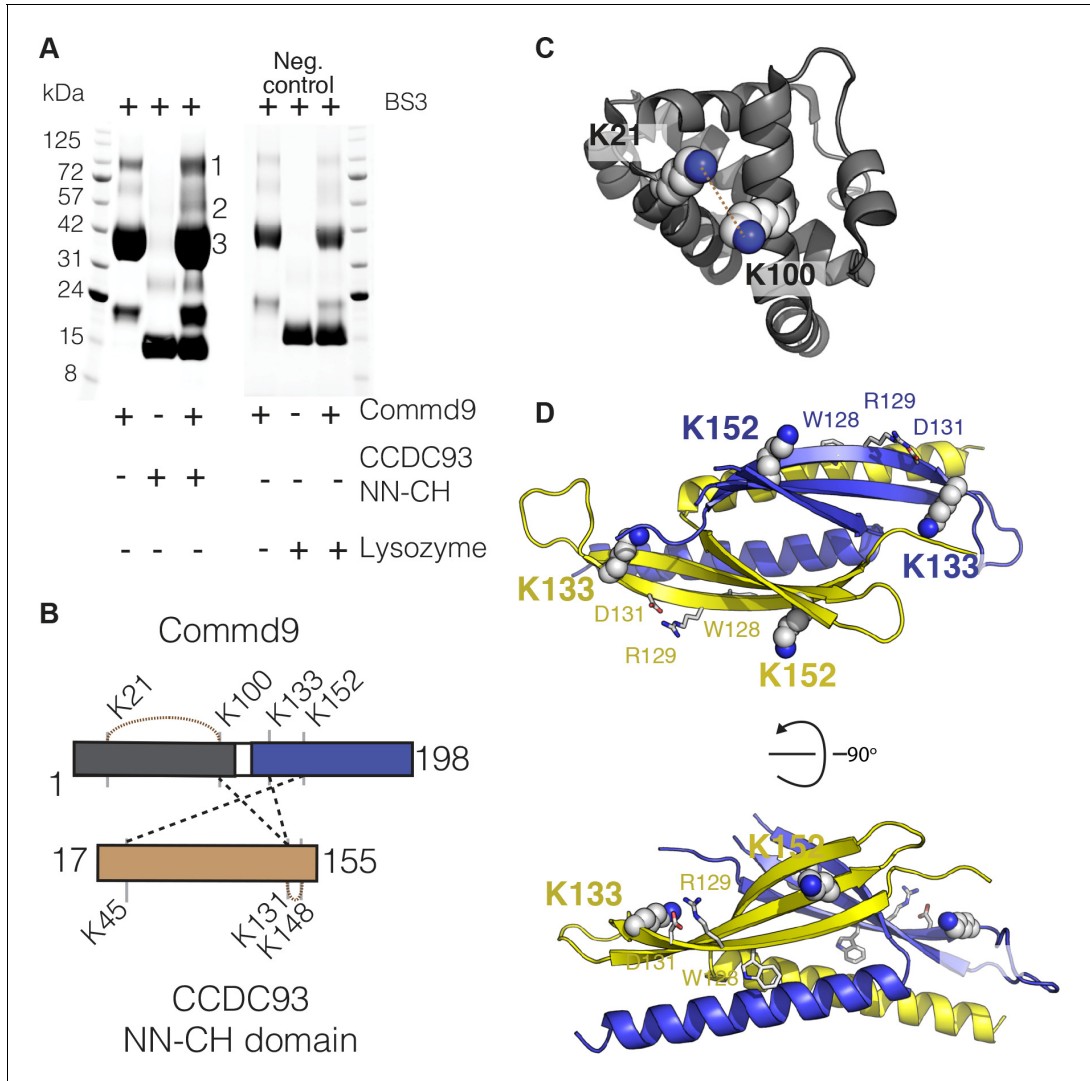

**Figure 7.** COMMD proteins bind to NN-CH domain of CCDC93 via the conserved WRVD motif in the COMM domain. (**A**) Cross-linking of Commd9, NN-CH domain of CCDC93 and the mixture of two with BS3 for 30 min at room temperature. Three distinguishable SDS-PAGE gel bands were excised in the complex mixture sample (indicated by the number 1, 2 and 3) for MS analysis. In parallel, a cross-linking reaction between Commd9 and lysozyme was performed under the same condition. (**B**) Cross-link map for Commd9 in complex with the NN-CH domain of CCDC93. Intermolecular and intramolecular cross-linked peptides are labelled as black and brown dot lines respectively. (**C** and **D**) Ribbon representation of the Commd9 HN (**C**) and (**D**) COMM domain structures with mapped intramolecular and intermolecular cross-linked lysine residues (in spheres). In the Commd9 COMM domain, K133 and K152 are part of a contiguous surface that includes the side-chains of the conserved residues [128]WRVD[131] (indicated with sticks).
DOI: https://doi.org/10.7554/eLife.35898.017

The following figure supplement is available for figure 7:

**Figure supplement 1.** Cross-linking of zfRetromer complex.
DOI: https://doi.org/10.7554/eLife.35898.018

observed, which were excised for MS analysis (*Figure 7A*). Lysozyme was used as a negative control, and a similar presence of bands 1 and 3 suggest they are likely to be cross-linked dimers and tetramers of Commd9 alone (*Figure 7A*). This observation was further supported by inspection the MS spectra and xQuest database (*Table 3*). Analyses of MS data from band 2 revealed 5 unique crosslinks, and 3 of these pairs connected Commd9 and NN-CH domain of CCDC93 (*Figure 7B* and *Table 3*). Mapping of these pairs onto the crystal structures of Commd9 showed that all three lysines (K100, K133 and K152) were located on the surface accessible to the solvent (*Figure 7C,D*). Notably, K133 and K152 are located on contiguous surfaces of the β-sheets of the COMM domains (*Figure 7D*), suggesting a likely binding surface for the CCDC93 NN-CH domain. This surface also includes the side-chains of the conserved residues [128]WRVD[131], with Trp128 in particular being strictly conserved across the entire Commd family (*Figure 2B*).

To minimize the possibility that the cross-linked peptides captured were non-specific interactions caused by BS3, we also performed a cross-linking reaction using the unrelated VPS26 – VPS29 – VPS35 retromer complex from zebrafish (hereafter designated as zfRetromer) as a positive control. Inspection of the xQuest database and the MS spectra reveal a total of 18 cross-linked peptides (*Table 4*). VPS26 and VPS29 mainly cross-linked to two opposite regions of VPS35, which is in good agreement with the known crystal structures of Retromer (*Figure 7—figure supplement 1*). This implies that the cross-links we observe between Commd9 and CCDC93 are specific.

## COMMD proteins associate non-specifically with negatively charged phospholipids

COMMD proteins are peripheral membrane proteins commonly associated with endosomal compartments (*Phillips-Krawczak et al., 2015*; *Bartuzi et al., 2016*; *Wan et al., 2015*; *McNally et al., 2017*; *Burkhead et al., 2009*; *Drévillon et al., 2011*). To examine if COMMD family proteins possess membrane-binding properties we performed qualitative liposome-pelleting assays with purified

**Table 3.** Identified cross-linked peptides of Commd9 and NN-CH domain of CCDC93.

| Predicted mass (Da) | Observed mass (Da) | Error (ppm) | xQuest score | Cross-linked peptides | |
|---|---|---|---|---|---|
| Sample 1: | | | | Commd9 - Commd9 | |
| 2751.419 | 2751.407 | 2.3 | 21.10 | VDIK(133)TSSDSISR-VDIK(133)TSSDSISR | |
| 2305.209 | 2305.206 | 3.0 | 20.35 | ASSK(21)DVVR-VDIK(133)TSSDSISR | |
| 4316.319 | 4316.306 | 3.2 | 18.78 | DLSSAEAILALFPENFHQNLK(95)NLLTK(100)IILEHVSTWR | |
| 3266.789 | 3266.782 | 1.8 | 18.67 | NLLTK(100)IILEHVSTWR-VDIK(133)TSSDSISR | |
| 2820.589 | 2820.579 | 2.4 | 17.28 | ASSK(21)DVVR-NLLTK(100)IILEHVSTWR | |
| 3648.889 | 3648.881 | 1.0 | 15.22 | LVDLDWRVDIK()TSSDSISR-VDIK(133)TSSDSISR | |
| 3202.679 | 3202.677 | 2.0 | 14.22 | ASSK(21)DVVR-LVDLDWRVDIK(133)TSSDSISR | |
| 2571.299 | 2571.288 | 4.1 | 12.78 | IQEDPSLCGDK(163)PSISAVTVELSK(175) | |
| 1859.009 | 1859.006 | 2.7 | 12.40 | ASSK(21)DVVR-ASSK(21)DVVR | |
| Sample 2: | | | | Commd9 - CCDC93 | |
| 2998.479 | 2998.476 | 1.5 | 16.86 | VDIK(133)TSSDSISR | AIETK(131)EEMGDYIR |
| 3513.859 | 3513.842 | 1.4 | 14.74 | NLLTK(100)IILEHVSTWR | AIETK(131)EEMGDYIR |
| 4772.479 | 4772.501 | 5.1 | 7.11 | MAVPTCLLQMK(152)IQEDPSLCGDKPSISAVTVELSK | IK(45)GLSPFDK |
| | | | | Commd9 - Commd9 | |
| 2820.589 | 2820.583 | 1.1 | 18.60 | ASSK(21)DVVR-NLLTK(100)IILEHVSTWR | |
| | | | | CCDC93 - CCDC93 | |
| 3569.679 | 3569.675 | 0.3 | 14.92 | AIETK(131)EEMGDYIR-SYSVSQFQK(148)TYSLPED | |
| Sample 3: | | | | Commd9 - Commd9 | |
| 2820.589 | 2820.599 | 4.6 | 20.02 | ASSK(21)DVVR-NLLTK(100)IILEHVSTWR | |
| 2305.209 | 2305.226 | 5.8 | 19.51 | ASSK(21)DVVR-VDIK(133)TSSDSISR | |

DOI: https://doi.org/10.7554/eLife.35898.019

**Table 4.** Identified cross-linked peptides of zfRetromer.

| Predicted mass (Da) | Observed mass (Da) | Error (ppm) | xQuest score | zfVPS26 | zfVPS29 | zfVPS35 |
|---|---|---|---|---|---|---|
| Intermolecular cross-linked peptides | | | | zfVPS26 | zfVPS29 | zfVPS35 |
| 2929.569 | 2929.550 | 8.0 | 13.20 | APEK(301)MR | | K(90)VADLYELVQYAGNIIPR |
| 1844.019 | 1844.040 | 9.4 | 10.12 | SKYHLK(188) | | NK(38)LMDALK |
| 1342.809 | 1342.805 | 4.9 | 9.99 | MRK(304)R | | K(127)DILK |
| 1756.069 | 1756.070 | 2.8 | 9.83 | | FK(23)KLLVPGK | WEKK(556) |
| 3809.959 | 3809.982 | 7.1 | 8.01 | | GDFDENLNYPEQK(73)VVTVGQFK | TQCALAASK(659)LLK |
| Intramolecular cross-linked peptides | | | | zfVPS26 – zfVPS26 | | |
| 1441.829 | 1441.819 | 3.3 | 28.12 | VNINVK(57)QTSK(61)R | | |
| 1630.869 | 1630.863 | 4.6 | 26.54 | TAELK(30)TEEGK(35)LEK | | |
| 1247.689 | 1247.679 | 2.1 | 26.15 | DVNK(266)K(267)FSVR | | |
| 2571.239 | 2571.229 | 4.3 | 22.06 | TEEGK(35)LEK(38)HYLFYDGESVSGK | | |
| 3241.639 | 3241.662 | 7.0 | 20.14 | K(25)TAELKTEEGK(35)LEKHYLFYDGESVSGK | | |
| 3626.859 | 3626.875 | 5.8 | 19.78 | K(214)EMTGIGPSTTTETETVAK-YFK(288)QQEIVLWR | | |
| 844.489 | 844.487 | 4.4 | 18.66 | K(25)TAELK(30) | | |
| 1014.549 | 1014.544 | 1.4 | 12.32 | K(297)APEK(301)MR | | |
| Intramolecular cross-linked peptides | | | | zfVPS29 - zfVPS29 | | |
| 1184.749 | 1184.736 | 5.9 | 18.59 | FK(23)KLLVPGK(30) | | |
| Intramolecular cross-linked peptides | | | | zfVPS35 – zfVPS35 | | |
| 1607.729 | 1607.726 | 3.3 | 22.38 | ENSSSDDK(552)WEKK(556) | | |
| 844.469 | 844.449 | 6.9 | 19.26 | EK(208)REK(211) | | |
| 2245.219 | 2245.215 | 5.1 | 18.55 | LLDEAVQAVK(24)VQSFQMK(31)R | | |
| 1255.699 | 1255.688 | 2.5 | 17.69 | LLK(662)K(663)PDQCR | | |

DOI: https://doi.org/10.7554/eLife.35898.020

COMMD proteins (*Figure 8A*, *Figure 8—figure supplement 1*). The assay shows that Commd1 associates relatively non-specifically with various negatively charged membranes, including PC/PE liposomes doped with different phosphoinositides, generic Folch lipids from brain extracts, and liposomes containing 30% phosphatidylinositolserine (PS). We observed a similar binding characteristic for Commd7, but Commd10 showed a relatively strong interaction with Folch liposomes as well as di- and tri- phosphorylated phosphoinositide species.

In contrast to the other family members, Commd9 appears to associate weakly if at all with the membranes tested (*Figure 8A*). A comparative analysis of the electrostatic surface of the COMM domain of Commd9 and a homology model of the COMM domain of Commd1 (constructed using the COMM domain of Commd9 as the template) shows that the Commd1 COMM domain exhibits a basic patch on its surface composed of solvent-exposed positively charged residues that are absent in Commd9 (*Figure 8B,C*). To test if this basic surface on Commd1 is involved in membrane recruitment, we made a triple mutant in the putative lipid-binding site (R133Q, H134A and K167A). In liposome-pelleting assays this mutant shows a drastic reduction in membrane interaction (*Figure 8A*).

Using BLiTz we next quantified the interactions of Commd1 and Commd9 with different phosphoinositide-containing membranes. While Commd1 possesses a higher level of binding response for phosphoinositides PI(3)$P$ and PI(4,5)$P_2$ compared to Folch and PS containing membranes, Commd9 showed little association with any of the lipids tested, in line with the liposome-pelleting assay (*Figure 8—figure supplement 2*). We performed a concentration-dependent binding series with a selection of liposomes and Commd1 to calculate the dissociation constants of these interactions. Commd1 binds to a variety of liposomes with similar affinities in the range of 1–10 μM, which is in agreement with the qualitative results from pelleting assays (*Figure 8D,E,F,G and H*). Confirming the importance of the basic patch on the Commd1 COMM domain, no interaction was observed

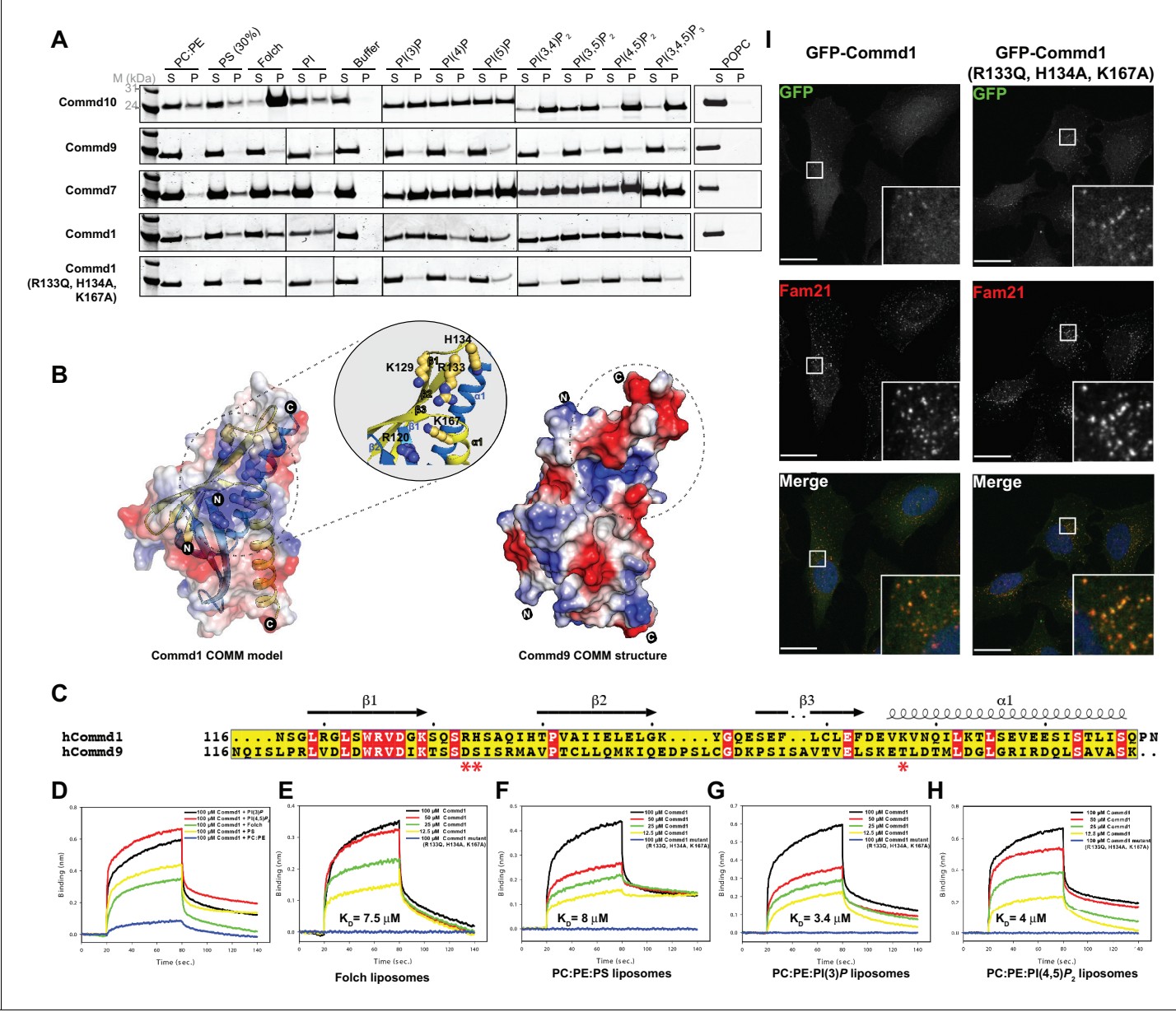

**Figure 8.** COMMD proteins bind membrane phospholipids but this is not required for endosomal recruitment. (**A**) Commd1, Commd1 (R133Q, H134A, K167A), Commd7, Commd9 and Commd10 were incubated with POPC (100%), POPC/POPE (90:10, molar ratio), POPC/POPE/POPS (60:10:30, molar ratio) and POPC/POPE/PIP (80:10:10, molar ratio) liposomes doped with different phosphoinositides to perform liposome pelleting assay by ultracentrifugation and subsequent protein content analysis of the supernatant (S) and pellet (P) fractions. (**B**) (Left) Homology model of Commd1 COMM domain shown as a transparent electrostatic surface overlaid with the ribbon representation, highlighting the residues crucial for forming the positively charged lipid-binding pocket. Inset shows the close up of the lipid-binding pocket. (Right) The electrostatic surface representation of Commd9 COMM domain, highlighting the absence of basic patch. (**C**) Sequence alignment of Commd1 and Commd9 COMM domains. Red asterisks mark the positively charged amino acids constituting the basic patch on Commd1, which are absent in Commd9. (**D–H**) Binding of liposomes containing various phosphoinositides to Commd1 measured at different concentrations and Commd1 (R133Q, H134A, K167A) using the BLiTz system. Binding kinetics was calculated using sigma-plot (Systat Software Inc.). (**I**) Lentiviral constructs of GFP-tagged Commd1 and Commd1 (R133Q, H134A, K167A) were transfected into HeLa cells and colocalisation with the Fam21 WASH subunit imaged by confocal immunofluorescence microscopy. The mutant Commd1 is still recruited to endosomes, presumably due to incorporation into heteromeric COMMD complexes and the CCC/Retreiver subunits.

DOI: https://doi.org/10.7554/eLife.35898.021

The following figure supplements are available for figure 8:

**Figure supplement 1.** Raw files of SDS-PAGE gels of liposome pelleting assay with COMMD proteins.

*Figure 8 continued on next page*

*Figure 8 continued*

DOI: https://doi.org/10.7554/eLife.35898.022

**Figure supplement 2.** Binding comparison of Commd1 and Commd9 with liposomes using BLiTz.

DOI: https://doi.org/10.7554/eLife.35898.023

with the Commd1 triple mutant. Altogether, the COMM domain of COMMD proteins appears to be a central hub for both protein-protein and protein-membrane interactions.

We next assessed the importance of the Commd1 membrane-binding surface for its cellular localization. GFP-tagged Commd1 wild type and mutant were expressed in HeLa cells following lentiviral transduction (*Figure 8I*). As seen previously, Commd1 is localized to endosomal puncta, and colocalises with the WASH complex subunit Fam21. Surprisingly however, the mutant Commd1 is also colocalised with Fam21 on endosomes, despite being defective for membrane binding in vitro. We propose therefore that this site on Commd1 may instead be important for non-specific membrane interactions and perhaps for the orientation of the CCC complex on endosomes, but in the context of the assembled complex with other COMMD proteins it is not required for specific recruitment to PtdIns3*P*-enriched endosomal compartments.

## Discussion

The COMMD proteins were recently identified as conserved and central components of the CCC complex (*Mallam and Marcotte, 2017*; *Wan et al., 2015*; *McNally et al., 2017*), a large endosome-associated assembly that regulates cell surface recycling of various transmembrane receptors (*McNally et al., 2017*; *Phillips-Krawczak et al., 2015*; *Bartuzi et al., 2016*; *Wan et al., 2015*). Despite the high degree of conservation of the COMMD proteins, and the CCC complex more generally, little is known about the structures or the stoichiometries of the component subunits, or how these proteins assemble together to control membrane recruitment and protein trafficking. In this study we provide the first insights into the architecture of the COMMD proteins, and the mechanisms that underpin the previously reported interactions between the various family members. COMMD proteins form obligatory dimers, and interact with each other in promiscuous homo- and heterodimeric arrangements via the C-terminal COMM domain. Furthermore this small domain is essential for pleiotropic interactions with both the CCC proteins CCDC93 and CCDC22, and with negatively charged phospholipid membranes. The solution structures of Commd1, Commd7 and Commd9 reveal a modular architecture with the α-helical N-terminal HN domains arranged as flexible appendages to the C-terminal COMM domain dimers. Previous studies had described the N-terminal region as a variable domain that could provide functional diversity to the COMMD proteins, due to the low level of sequence homology in this region across the family. Our biophysical, structural and bioinformatics analyses however show that the HN domain is a structurally conserved feature shared by all COMMD family members.

The architecture of the COMMD proteins revealed here bears a remarkable resemblance to the orthologous CT584 and Cpn0803, *Chlamydia*-specific proteins from *C. trachomatis* and *C. pneumonia* respectively. The function of these proteins is unknown, although there is evidence that Cpn0803 can interact with phospholipids as well as components of the Type III secretion system responsible for injection of bacterial effectors into the host cytoplasm (*Stone et al., 2012*). What does the homology of COMMD proteins to these chlamydial proteins imply? Chlamydial species are intracellular pathogens that reside inside a membrane vacuole called the inclusion. During their life cycle they secrete many effectors that can hijack the intracellular trafficking and signalling machinery of the host cell to promote survival (*Elwell et al., 2016*; *Mirrashidi et al., 2015*; *Fischer et al., 2017*; *Pruneda et al., 2016*). We speculate therefore that the CT584/Cpn0803 proteins may be secreted effectors with a potential to mimic or modify COMMD protein functions, although this remains to be confirmed. The evolutionary relationship of the COMMD proteins to the chlamydial homologues is also an interesting question. Several chlamydial proteins are believed to have evolved through horizontal gene transfer from eukaryotic hosts, such as SWIB domain-containing proteins and Swi/Snf2 helicases (*Bastidas and Valdivia, 2016*; *Stephens et al., 1998*), and it is possible that the CT584/Cpn0803 genes have been acquired in a similar fashion. Both CT584 and Cpn0803 form hexameric structures composed of a trimer of dimers, with the N-terminal domains providing the primary

interface for trimer formation (*Barta et al., 2013*; *Stone et al., 2012*) (*Figure 4—figure supplement 4E*). While full-length recombinant Commd1, Commd7 and Commd9 proteins analysed here form homodimers, it is tempting to speculate that the N-terminal domains of COMMD proteins will contribute to formation of higher-order heteromeric assemblies that are present in the 600 kDa CCC complexes isolated from cultured cells (*Wan et al., 2015*), the stoichiometries and structures of which remain to be determined.

Although the COMMD proteins play a central role in endosomal membrane trafficking as components of the CCC complex, they have also been implicated in a number of other cellular processes. The gene encoding Commd1 (originally called Murr1) was originally identified in dogs with copper toxicosis, which has been mechanistically linked to interactions with the Wilson disease ATPase protein ATP7B (*Tao et al., 2003*). Most prominently, Commd1 has been shown to be a potent inhibitor of the transcription factor NF-κB, a master regulator of inflammation, and other COMMD proteins also display similar activities (*Maine and Burstein, 2007*; *de Bie et al., 2006*; *Ganesh et al., 2003*; *Bartuzi et al., 2013*). Commd1 acts downstream of the inhibitory IκB kinase (*Ganesh et al., 2003*), and is believed to associate with NF-κB subunits at chromosomal loci and promote NF-κB ubiquitination by Cullin family ubiquitin ligases for degradation (*Maine et al., 2007*; *Mao et al., 2011*). A similar function has been proposed for Commd1 regulation of the HIF-1α transcription factor (*van de Sluis et al., 2007*, *2010*, *2009*). The COMM domain fold is structurally related to the repeat domains found in the PUR (purine-rich element binding protein) family protein Pur-α (*Graebsch et al., 2009*; *Weber et al., 2016*). Pur-α contains three such repeat domains, the first two of which form an intramolecular 'dimer' while the third forms an intermolecular dimer (*Weber et al., 2016*) that both resemble the dimeric COMM domain topology. Pur-α is a nucleic acid binding protein that plays important roles in the transcription of neuronal genes (*Gallia et al., 2000*; *White et al., 2009*), and is associated with GGGGCC-containing inclusions in ALS-FTD patients (*Xu et al., 2013*). The similarity of the COMM domain with Pur-α suggests the intriguing possibility that COMMD proteins could also interact with DNA and/or RNA to regulate transcription, although it should be noted that the specific DNA binding site in Pur-α (*Weber et al., 2016*) is not conserved in the COMMD proteins (*Figure 4—figure supplement 3A*).

The CCC complex and associated COMMD proteins are predominantly localised to early endosomal membranes (*Phillips-Krawczak et al., 2015*; *Bartuzi et al., 2016*; *Wan et al., 2015*; *McNally et al., 2017*; *Burkhead et al., 2009*; *Drévillon et al., 2011*). The mechanism of membrane recruitment of the complex however is unknown. Most intracellular trafficking complexes rely on binding to proteins such as the Rab GTPases and membrane lipids such as phosphoinositides for their spatio-temporal recruitment to specific compartments (*Cullen, 2011*). COMMD proteins show significant but non-preferential binding to a variety of negatively charged lipids, and these interactions appear to be maintained by the COMM domain. Our data also highlights that the HN domain is dispensable for membrane association, which is in line with the work of Burkhead et al. (*Burkhead et al., 2009*). The membrane-binding surface of Commd1 in vitro is composed of residues from both molecules of the homodimer, but interestingly this does not appear essential for endosomal localisation of Commd1 in cells. This suggests that the ability of COMMD proteins to associate with membranes is only one part of a more complicated array of interactions controlling specific endosomal membrane recruitment, likely involving different lipid binding properties of heterodimeric COMMD proteins and other CCC complex subunits. Supporting a role for other subunits and complexes in specific endosomal localisation of the CCC complex, the depletion of CCDC93 or the WASH complex subunit Fam21 both lead to a loss of Commd1 from endosomes (*Phillips-Krawczak et al., 2015*), while the depletion of CCDC22 causes a redistribution of Commd1 and Commd10 (*Starokadomskyy et al., 2013*).

The ability to self-assemble and form heteromeric complexes is a core property of the COMMD proteins identified in some of the very first studies (*Burstein et al., 2005*). Our work provides a clear structural explanation for how homo- and heterodimers are formed by the different family members, and begins to suggest mechanisms for how different domains could contribute to the formation of larger assemblies, how they associate with biological membranes, and how they become incorporated into the CCC complex through interactions with the CCDC proteins. As COMMD-containing complexes emerge as key regulators of cellular trafficking, signalling and transcription, the details of these molecular interactions and the systems that regulate them remain outstanding questions to be answered.

## Materials and methods

### Antibodies and phospholipids

The monoclonal mouse anti-His antibody was purchased from Genscript Corporation (catalog no. A00186, 1:5000). Goat anti-mouse IgG HRP was from Life Technologies (Catalog no. A16072, 1:1000). POPC (1-palmitoyl-2oleoyl-sn-glycero-3-phosphocholine) (catalog no. 850475P), POPE (1-palmitoyl-2oleoyl-sn-glycero-3-phosphoethanolamine) (catalog no. 850757P), DOPS (1,2-dioleoyl-snglycero-3-phosphoserine) (catalog no. 850150P) and biotinylated POPE (1-palmitoyl-2oleoyl-sn-glycero-3-phosphoethanolamine-N-(biotinyl)) (catalog no. 870285P) were purchased from Avanti Polar Lipids. PI (1,2-dioleoyl-sn-glycero-3-phospho-(1'-myo-inositol)) (catalog no. P-0016), PI(3)P (1,2-dioctanoyl-sn-glycero-3-(phosphoinositol-3-phosphate)) (catalog no. P-3016), PI(4)P (1,2-dioleoyl-sn-glycero-3-phospho-(1'-myo-inositol-4'-phosphate)) (catalog no. P-4016), PI(5)P (1,2-dioleoyl-sn-glycero-3-phospho-(1'-myo-inositol-5'-phosphate)) (catalog no. P-5016), PI(3,4)$P_2$ (1,2-dioctanoyl-sn-glycero-3-phospho-(1'-myo-inositol-3',4'-bisphosphate)) (catalog no. P-3416), PI(3,5)P2 (1,2-dioctanoyl-sn-glycero-3-phospho-(1'-myo-inositol-3',5'-bisphosphate)) (catalog no. P-3516), PI(4,5)$P_2$ (1,2-dioctanoyl-sn-glycero-3-phospho-(1'-myo-inositol-4',5'-bisphosphate)) (catalog no. P-4516) and PI(3,4,5)$P_3$ (1,2-dioctanoyl-sn-glycero-3-phospho-(1'-myo-inositol-3',4',5'-trisphosphate)) (catalog no. P-3916) lipids were obtained from Avanti Polar Lipids, Inc.

### Molecular biology and cloning

All the constructs cloned into bacterial expression plasmids are listed in *Figure 1—figure supplement 1*. Briefly, DNA encoding full-length human Commd proteins and CCDC22 and, 93 was cloned into the pGEX-4T-2 plasmid for expression as N-terminal GST-tagged fusion proteins. Full length Commd1, 7 and 9 were also cloned into the pET30b(+) vector with a C-terminal His6 tag by Genscript Corporation. Commd1 HN domain (1-114), Commd1 COMM (116-190), Commd9 HN domain (1- 116), Commd9 COMM (115-198), CCDC22_N (1-139), CCDC22_C (194-627), CCDC93_N (17-155) and CCDC93_C (239-630) were artificially synthesized by Genscript Corporation and cloned into both pGEX-6P-1 and pET30b(+) plasmid as N-terminal GST tagged and C-terminal His6 tagged fusion proteins respectively. Genscript Corporation generated all the mutants used in this study. The construct used in crystallization of Commd9 HN domain (1-117) was cloned into pDEST17 with an N-terminal His6 tag.

### Recombinant protein expression and purification

The bacterial expression plasmids were transformed into *Escherichia coli* BL21-CodonPlus (DE3)-RIPL competent cells (Agilent). The bacterial cultures were grown in LB until OD$_{600nm}$ reached 0.6. The cultures were cooled to 18°C before inducing protein expression by adding 0.5 mM isopropylthio-β-galactoside (IPTG) and allowed to grow for 16 h (*Ghai et al., 2011*). The cells were harvested by centrifugation at 6000 × g for 5 min at 4°C and the harvested cell pellet was resuspended in lysis buffer [20 mM Tris (pH 8.0), 500 mM NaCl, 10% glycerol, 0.2% IGEPAL, 50 μg/mL benzamidine, 100 units DNaseI, and 1 mM β-mercaptoethanol]. Cells expressing His6-fused proteins were resuspended in lysis buffer supplemented with 20 mM imidazole (pH 8.0). The cells were lysed by mechanical disruption at 30 kpsi using a Constant systems cell disrupter. The lysate was clarified by centrifugation at 50,000 × g for 30 min at 4°C. Proteins were purified using affinity chromatography from the clarified lysate.

His-tagged proteins were purified on a nickel-NTA (Clonetech) gravity column and eluted with 500 mM imidazole in buffer containing 150 mM NaCl, 20 mM Tris (pH 8.0) and 1 mM β-mercaptoethanol. GST-tagged proteins were purified on a glutathione-Sepharose (GE healthcare) gravity column and eluted with 10 mM glutathione, 150 mM NaCl, 20 mM Tris (pH 8.0) and 1 mM β-mercaptoethanol or the GST tag was cleaved with the addition of thrombin or precision protease cleavage on to the beads with overnight incubation at room temperature. Finally, proteins were subjected to size exclusion chromatography using a superdex-200 16/60 Hiload column or superdex-75 16/60 Hiload column attached to an AKTA pure (GE Healthcare).

The Commd5-Commd10 complexes were reconstituted by co-transformation of GST-Commd5 or GST-tagged COMM domain of Commd5 with Commd10-His or His-tagged COMM domain of Commd10 into *Escherichia coli* BL21 (DE3) competent cells (Agilent). The cells were cultured,

harvested and lysed as described above. The complexes were first purified using glutathione sepharose beads and the GST tag was cleaved using prescission protease. The eluted fractions were then mixed with equilibrated TALON beads to obtain a 1:1 stoichiometric complex. The complexes were eluted by supplementation of 200 mM Imidazole (pH 8.0) in the wash buffer. The eluted proteins were subjected to size exclusion chromatography using a superdex-200 16/60 Hiload column.

For crystallization, SAXS, and MALLS experiments, proteins were buffer exchanged into 10 mM Tris (pH 8.0), 100 mM NaCl and 2 mM DTT using SEC. For the structure determination of Commd9 COMM domain, the protein was labeled with selenomethionine using the method described by Van Duyne et al (*Van Duyne et al., 1993*). zfVPS26, zfVSP29 and zfVPS35 were expressed separately in *E. coli* BL21 (DE3) cells grown in LB at 37°C, and protein expression was induced at an OD600 of 0.7–0.8 by the addition of 1 mM IPTG. Cells were harvested after 18 hr of growth at 18°C. Cell pellets of overexpressed zfVPS26, zfVSP29 and zfVPS35 were mixed and purified using standard metal affinity, glutathione affinity and size-exclusion chromatography techniques to obtain the purified retromer complex. The size-exclusion buffer contains 50 mM HEPES (pH 7.5), 150 mM NaCl, 2 mM DTT.

## Multi-angle laser light scattering

The molecular mass of the COMMD proteins was determined by size exclusion chromatography on an AKTA pure (GE Healthcare) connected to a multi angle laser light scattering and, differential refractive index (RI) detector. The protein samples were gel-filtered in a buffer containing 25 mM Tris (pH 8.0), 300 mM NaCl and 2 mM DTT that had been filtered (0.22 µm) and degassed. Measurements of full-length COMMD proteins were made using a superdex-200 increase 5/150 column (GE Healthcare) at a flow rate of 0.25 ml/min with in-line UV, MALLS, and RI detectros (Dawn Heleos II and Optilab reX, respectively, Wyatt Technology Corp) for $M_W$ characterization. Measurements of the COMMD COMM and HN domains were made using a superdex-75 5/150 column (GE Healthcare) at a flow rate of 0.25 ml/min. UV, MALLS and RI data were collected and analysed using the ASTRA$^{TM}$ software (Wyatt Technology) (*Folta-Stogniew, 2006*) to compute the molecular mass.

## Crystallisation, data collection and structure determination

The Commd9 COMM domain was buffer-exchanged into 10 mM Tris (pH 8.0), 100 mM NaCl, 2 mM DTT, and concentrated to 8 mg/ml for crystallisation at 20°C. The protein was supplemented with 10 mM DTT before setting up hanging-drop crystallization screens using a mosquito liquid handling robot (TTP LabTech). Commd9 COMM was crystallised in 0.1 M HEPES (pH 7.0), 6% Jeffamine M-600.

In the case of the Commd9 HN domain, protein was buffer exchanged into 50 mM HEPES (pH 8.0), 200 mM NaCl, 1 mM tris (2-carboxyethyl) phosphine (TCEP) and initial crystallisation screens were set up at 12 mg/ml at 18 °C. Crystals of the HN domain were obtained in 0.2 M citric acid (pH 4.9), 28% MME-PEG5000 in a hanging drop setup respectively. Data were collected at the Australian Synchrotron MX1 and MX2 Beamlines. iMOSFLM (*Battye et al., 2011*) was used to integrate the data, and AIMLESS (*Evans and Murshudov, 2013*) was used for data scaling in the CCP4 suite (*Winn et al., 2011*). The Commd9 COMM domain structure was solved using single anamolous dispersion (SAD), and the phases were calculated using the peak wavelength data of selenium with AUTOSOL using the PHENIX suite (*Adams et al., 2010*; *Terwilliger et al., 2009*). The solution from AUTOSOL was built using autobuild (*Terwilliger et al., 2008*) and the resulting model was rebuilt with COOT (*Emsley and Cowtan, 2004*) followed by repeated refinement runs and model building with PHENIX (*Adams et al., 2010*) and COOT (*Emsley and Cowtan, 2004*). The Commd9 HN domain structure was determined using multiwavelength anamolous dispersion (MAD) and the phases were obtained using the program SOLVE. Model building and refinement was done using COOT and PHENIX refine.

## Small angle X-ray scattering

In line SEC-SAXS measurements on homogeneous protein samples (assessed using MALLS) were performed at the SAXS/WAXS beamline at the Australian Synchrotron using a superdex-200 increase 5/150 column (GE Healthcare), and Pilatus 1M detector (Dectris). The scattering data were measured in a q range of 0.011 to 0.4 Å at 12 keV using a 1.6 m camera length. Samples were loaded on to

the size exclusion column that was equilibrated with 10 mM Tris (pH 8.0), 100 mM NaCl, 2 mM DTT and 5% Glycerol. Data reduction was performed using the ScatterBrain program (written and provided by the Australian Synchrotron; available at http://www.synchrotron.org.au). The buffer frames were averaged after assessing the statistical equivalence using CorMap p values with a significance threshold ($\alpha$) of 0.01. The averaged buffer scattering was subtracted from statistically similar data from Commd proteins elution peak. $R_g$ was evaluated using the Guinier approximation and was found to be consistent under the elution peak. Primary data processing was performed in Primus using the ATSAS suite (version 2.6) (*Petoukhov et al., 2012*). Pair distance distribution *P(r)* of Commd proteins was determined using GNOM. $C_2$ symmetry was assumed in generating low-resolution three-dimensional *Ab initio* envelopes using the program GASBOR (*Kozin and Svergun, 2001*). DAMAVER (*Volkov and Svergun, 2003*) was used to average the 20 independent models generated by GASBOR. Rigid body modeling was performed using SASREF (*Petoukhov and Svergun, 2005*) and the partial scattering amplitudes were calculated with CRYSOL (*Svergun et al., 1995*). The *ab initio* models were superimposed on to the rigid body modeled structures using SUPCOMB (*Kozin and Svergun, 2001*).

## Co-expression GST pull-downs

BL21-CodonPlus (DE3)-RIPL competent cells were co-transformed with either His6-tagged Commd1, Commd1 COMM domain, Commd1 HN domain, Commd9, Commd9 COMM domain or Commd10 with each of the GST-tagged proteins of interest (Commd1-10, CCDC22, CCDC22_N (1-139), CCDC22_C (194-627), CCDC93_N (17-155), CCDC93_C (239-630) and empty pGEX4T-2 vector (expressing GST). Transformants were selected via overnight growth using triple antibiotic agar plates. A single colony was picked to initiate the culture and proteins were co-expressed using the standard protein expression protocol as described above. Proteins were purified by affinity chromatography using glutathione sepharose beads (GE healthcare) and SDS-PAGE was run to visualize GST-tagged bait proteins. Binding of His-tagged proteins (prey) to GST-tagged (bait) was observed by Western blotting using mouse anti-His antibody (Genscript). Genscript generated all the mutants used in this study.

## GST pull downs

1 nmol GST-tagged CCDC22_N (1-139) and CCDC93_N (17-155) were mixed with 1 nmol of His-tagged Commd1 and Commd9, and COMM domains of Commd1 and Commd9 and Commd1 HN domain, for 1 hr at 4°C. Protein mixture was then centrifuged at high speed to remove any precipitated proteins. The supernatant was then added to pre-equilibrated (20 mM Tris (pH 8.0), 300 mM NaCl, 1 mM DTT) glutathione sepharose and allowed to mix for a further 30 min at 4°C. Beads were washed five times in the above buffer supplemented with 0.5% triton X100 (Sigma Aldrich). Bound proteins were analysed by Western blots using mouse anti-His antibody (Genscript).

## Chemical Cross-linking coupled with mass spectrometry

For cross-linking, the purified full-length Commd9 and NN-CH domain of CCDC93 mixture at 50 µM in 50 mM Hepes (pH 7.5), 150 mM NaCl were incubated with 100 molar excess of BS3-d0 cross-linker (Sigma-Aldrich) for 30 min at room temperature. The reaction was quenched by addition of 100 mM Tris-HCl (pH 8.5), and the cross-linked products were analysed by SDS-PAGE and subjected to MS analysis. A negative control cross-linking reaction was performed between the full-length Commd9 and lysozyme using the same condition described above. For the positive control cross-linking reaction, the purified retromer complex at 15 µM was reacted with 100 molar excess of BS3-d0 cross-linker. BS3-d0 was purchased from Sigma Aldrich (catalog no. S5799). The gel band that corresponds to the molecular weight of monomeric retromer complex was subjected to MS analysis. The bands from the SDS-PAGE gels were excised and reduced with dithioerythritol followed by alkylation with iodoacetamide. Alkylated samples were digested with trypsin (Promega) in 50 mM ammonium bicarbonate pH 8.0 overnight using an enzyme-to-substrate ratio of 1:100 (w/w) at 37°C. The digested samples were extracted using extraction buffer containing 5% formic acid and 50% acetonitrile followed by sonication for 1 min. The supernatant was then dried down in a vacuum centrifuge and redissolved in 0.1% formic acid prior to analyse by LC-MS/MS. The extracted peptides were analysed by uHPLC-MS/MS on an Eksigent, Ekspert nano LC400 uHPLC (SCIEX, Canada)

coupled to a Triple Tof 6600 mass spectrometer (SCIEX, Canada) equipped with a duo microelectrospray ion source. In brief, samples were injected onto a 300 μm x 150 mm ChromXP C18 CL 3 μm column (SCIEX, Canada) at 5 μl/min. The bound peptides were eluted with a gradient using solvent containing 0.1% formic acid in acetonitrile. 250 ms full scan TOF-MS data was acquired followed by up to 30 50 ms full scan product ion data in an Information Dependant Acquisition, IDA, mode. TOFMS data was acquired over the mass range 350–2000 and for product ion ms/ms 100–1600. Ions observed in the TOF-MS scan exceeding a threshold of 100 counts and a charge state of +2 to +5 were set to trigger the acquisition of product ion, ms/ms spectra of the resultant 30 most intense ions. Acquisition of all MS/MS samples was performed using Analyst TF 1.7 software (SCIEX, Canada). Inspection of the raw MS data was done using ProteinPilot software (SCIEX, Canada). The assignment of cross-linked peptides was made based on xQuest database search engine (*Rinner et al., 2008*). Trypsin was set as the enzyme used for digestion during sample preparation with an MS1 tolerance of 10 ppm and MS2 tolerance of 0.2 m/z.

## Liposome preparation

All the phosphoinositides were protonated prior to usage. In brief, powdered lipids were resuspended in chloroform ($CHCl_3$) and dried under argon. Dried lipids were then left in a desiccator for 1 hr to remove any remaining moisture. Dried lipids were resuspended in $CHCl_3$:Methanol (MeOH):1N hydrochloric acid in a 2:1:0.01 molar ratio, lipids were dried once again and allowed to desiccate. Lipids were then resuspended in $CHCl_3$:MeOH in a 3:1 ratio dried once again under argon. Finally, dried lipids were resuspended in $CHCl_3$ and stored at −20°C.

Lipid stock solutions were mixed to the desired molar ratios and dried under argon. To prepare control liposomes POPC and POPE were mixed in a 90:10 molar ratio, for BLiTz experiments liposomes were doped with 0.5% biotinylated POPE. Liposomes containing phosphoinositides were prepared by mixing POPC, POPE and PIPs in a 80:10:10 molar ratio respectively. 30% POPS was used for POPC:POPE:POPS. Dried lipids were hydrated in 25 mM HEPES (pH 7.2), and 220 mM sucrose to obtain a suspension of multilamellar liposomes containing sucrose. This solution was then freeze-thawed five times to produce unilamellar liposomes. Liposomes were then diluted 1:5 in 25 mM HEPES (pH 7.2), and 125 mM NaCl solution. The solution was then centrifuged at 250,000 g to remove sucrose from the medium and maintain osmolarity. The pelleted liposomes were resuspended in 25 mM HEPES (pH 7.2), and 125 mM NaCl solution to the desired concentration of 0.5 mM. All liposomes were used within 1 day of preparation.

## Liposome pelleting

10 μM of the protein of interest was added to a final volume of 200 μl of the liposome solution. This solution was left at room temperature for 25 min to allow for protein-liposome interaction. After incubation, the solution was centrifuged at 400,000 g for 30 min. Supernatant and pellet fractions were separated and the pellet was resuspended in 200 μl of 25 mM HEPES (pH 7.2), and 125 mM NaCl, samples were then collected for analysis on a precast 4–12% bis-tris gel (Novex) by coomassie staining.

## Biophysical interaction using Bio-layer interferometry (BLiTz)

Protein-lipid and protein-protein interactions were determined using the bio-layer interferometry from the BLiTz system. Protein-lipid interactions were observed by immobilizing 500 μM of biotinylated liposomes on a streptavidin biosensor. After immobilization, the sensor was washed with buffer containing 10 mM Tris (pH 8.0), 150 mM NaCl and 0.1% BSA to prevent non-specific association. Increasing concentrations (12.5, 25, 50 and 100 μM) of protein were added to the sensor and the change in binding (nm) was measured. Proteins were then allowed to disassociate from the probe in the buffer previously mentioned. The kinetics of the protein-protein interactions were determined in the same fashion using 5 μM of His-tagged COMMD1 COMM domain immobilized on a nickel-NTA probe and increasing protein concentrations of 62.5, 125, 250 and 500 μM. The data was processed and plotted using the Sigmaplot package (Systat Software Inc.).

## Cell culture

RPE1, HEK293T cells were maintained in DMEM (D5796; Sigma-Aldrich) plus 10% fetal calf serum (F7524; Sigma-Aldrich) under standard conditions. These cell lines were obtained from America Type Culture Collection (ATCC). Parental and stable cells lines were negative for mycoplasma by DAPI staining, and authenticated by STR profiling. Lentivirus particles for producing stably expressing cell lines were generated in HEK293T cells using the pXLG3 vector to carry the GFP tagged Commd1 WT and Commd1 (R133Q, H134A, K167A). Cells were transfected with DNA using polyethylenimine (Sigma-Aldrich). Virus was harvested from the growth media 72 hr post transfection.

For stable transduction with lentivirus, cells were seeded at 75,000 per well in six well plates. The cells were then incubated under normal conditions with titrations of viral supernatant for 72 hr. Cells were then passaged and expression of the GFP tagged protein of interest assessed by western analysis. Cell lines that displayed similar expression levels were selected for comparison and those closest to endogenous levels of protein.

## Immunofluorescence

RPE1 cells grown on 13 mm coverslips were washed with PBS before being fixed in ice cold 4% formaldehyde in PBS for 25 min. Cells were permeabilised in 0.1% Triton X-100 (Sigma) for 6 min. The cells were then blocked with 1% bovine serum albumin (BSA) in 0.01% Triton for 15 min at room temperature. Primary antibodies were diluted in 1% BSA and samples were incubated for 1 hr at room temperature. The samples were then incubated with Alexa Fluor conjugated secondary antibody and 0.2 μM DAPI for 30 min at room temperature. Coverslips were mounted in Mowiol-DABCO mounting medium (Sigma). Cells were visualised using a Leica TCS SP5 X confocal microscope (Leica Biosystems).

## Data availablity

Coordinates and structure factors for the COMM and HN domain of Commd9 have been deposited at the Protein Data Bank (PDB) with accession codes 6BP6 (COMM domain Commd9) and 4OE9 (HN domain of Commd9).

## Acknowledgements

We thank the University of Queensland Remote Operation Crystallisation and X-ray (UQ ROCX) facility and the staff for their support with crystallisation experiments; the staff of the Australian Synchrotron for assistance with X-ray diffraction; and SAXS experiments. We would also like to thank Willa Huston (UTS) for helpful discussions regarding chlamydial protein functions, and Alun Jones for his assistance with mass spectrometry data collection. This work was supported by funds from the National Health and Medical Research Council (NHMRC)-Australian Research Council (ARC) Dementia Research Development fellowship (1097185) and University of Queensland early career researcher grant (2016003796) to RG, and ARC (DP160101743), and NHMRC (APP1058734) to BMC. BMC is supported by NHMRC career development fellowship (APP1061574). Medical Research Council (MR/K018299/1 and MR/P018807/1) and the Wellcome Trust (089928 and 104568) grants supported PJC. JSL was supported by the Marsden fund administered by the Royal Society of New Zealand.

## Additional information

### Funding

| Funder | Grant reference number | Author |
|---|---|---|
| Royal Society of New Zealand | Marsden fund: UOO-04-09 | Manuela K Hospenthal<br>Dion J Celligoi<br>Fiona J McDonald<br>J Shaun Lott |
| Wellcome | 089928 | Peter J Cullen |
| Medical Research Council | MR/K018299/1 | Peter J Cullen |
| Wellcome | 104568 | Peter J Cullen |

| Medical Research Council | MR/P018807/1 | Peter J Cullen |
| Australian Research Council | DP160101743 | Brett M Collins |
| National Health and Medical Research Council | APP1058734 | Brett M Collins |
| National Health and Medical Research Council | APP1061574 | Brett M Collins |
| National Health and Medical Research Council | 1097185 | Rajesh Ghai |
| University of Queensland | Early career researcher grant: 2016003796 | Rajesh Ghai |

The funders had no role in study design, data collection and interpretation, or the decision to submit the work for publication.

### Author contributions

Michael D Healy, Formal analysis, Investigation, Methodology, Writing—original draft, Writing—review and editing; Manuela K Hospenthal, Mintu Chandra, Formal analysis, Investigation, Methodology; Ryan J Hall, Investigation; Molly Chilton, Vikas Tillu, Investigation, Methodology; Kai-En Chen, Data curation; Dion J Celligoi, Investigation, Methodology, Writing—review and editing; Fiona J McDonald, Peter J Cullen, J Shaun Lott, Formal analysis, Investigation, Methodology, Writing—review and editing; Brett M Collins, Conceptualization, Supervision, Funding acquisition, Investigation, Project administration, Writing—review and editing; Rajesh Ghai, Conceptualization, Data curation, Formal analysis, Supervision, Funding acquisition, Investigation, Methodology, Writing—original draft, Writing—review and editing

### Author ORCIDs

Michael D Healy (iD) http://orcid.org/0000-0003-2924-9179
Ryan J Hall (iD) http://orcid.org/0000-0002-8543-0370
Molly Chilton (iD) http://orcid.org/0000-0003-2238-9822
Vikas Tillu (iD) http://orcid.org/0000-0002-1034-9543
Kai-En Chen (iD) http://orcid.org/0000-0003-1106-1629
J Shaun Lott (iD) https://orcid.org/0000-0003-3660-452X
Brett M Collins (iD) http://orcid.org/0000-0002-6070-3774
Rajesh Ghai (iD) http://orcid.org/0000-0002-0919-0934

### Decision letter and Author response

Decision letter https://doi.org/10.7554/eLife.35898.031
Author response https://doi.org/10.7554/eLife.35898.032

# Additional files

### Supplementary files

• Transparent reporting form
DOI: https://doi.org/10.7554/eLife.35898.024

### Data availability

The raw biochemical data generated in this study is included in the supporting files. Diffraction data has been deposited in PDB and accession codes are provided in the manuscript.

The following datasets were generated:

| Author(s) | Year | Dataset title | Dataset URL | Database, license, and accessibility information |
|---|---|---|---|---|
| Healy MD, Chandra M, Collins BM, Ghai R | 2018 | Crystal structure of Commd9 COMM domain | http://www.rcsb.org/structure/6BP6 | Publicly available at the RCSB Protein Data Bank (accession |

| | | | | no: 6BP6) |
|---|---|---|---|---|
| Hospenthal M, Celligoi D, Lott JS | 2015 | The crystal structure of the n-terminal domain of COMMD9 | https://www.rcsb.org/structure/4OE9 | Publicly available at the RCSB Protein Data Bank (accession no: 4OE9) |

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
