## [Decision Letter]

Thank you for submitting your article "Structural insights into the architecture and membrane interactions of the conserved COMMD proteins" for consideration by *eLife*. Your article has been reviewed by three peer reviewers, and the evaluation has been overseen by a Reviewing Editor and Vivek Malhotra as the Senior Editor.

The reviewers have discussed the reviews with one another and the Reviewing Editor has drafted this decision to help you prepare a revised submission.

The reviewers are in broad agreement that the COMMD proteins have an important, if enigmatic, role in endosome function and that insight into their structure will help underpin studies aimed an better understanding this role. However, there is a concern that the biological significance of the structural data presented is not fully established. Such insight would be expected for a broad interest journal such as *eLife*. The reviewers have suggested that this could be addressed by using the structure to identify COMMD residues that are involved in binding CCDC22 and/or CCDC93.

If you feel that you can add this further experimental work, or carry out related experiments to address the same point, then we would be happy to consider a revised version. If you submit a revision, please tend to other aspects of the experimental work that need attention.

Summary:

COMMD proteins comprise a highly conserved eukaryotic protein family. Its 10 members in human are implicated in cellular trafficking and the regulation of transcription. The proteins consists of an N-terminal calponin homology-like domain (HN) and a C-terminal COMM domain. To understand the molecular basis for its function, the authors of this study determined the crystal structure of the isolated COMM and HN domain of COMMD9. They found that COMM domains form homo- and heteromeric assemblies via a highly conserved interface, whereas the HN domain is monomeric. SAXS measurements with the full-length constructs of Commd1, Commd7 and Commd9 were used to extract information on the quaternary assembly. Using pull-down studies, gel filtration analysis and Blitz measurements, they further characterized heteroassembly of COMMDs within the family and with the CCC components CCDC22 and 93. Finally, they found unspecific membrane-binding of Commd 10 to liposomes of various compositions. Mutations of a basic patch in Commd10 reduced membrane binding but not its cellular localization on endosomal structures.

Essential revisions:

1) COMMD homo- and heterodimers:

Homo- and heterodimerization seems to be promiscuous among Commd proteins. While Commd1, Commd7 and Commd9 behave as 1/1 homodimers (Figure 1A), the Commd5-Commd10 complex behaves as a 2/2 heterotetramer (Figure 7E). From Figure 7A, it seems that by co-expression in *E. coli*, Commd10 binds strongly to Commd2. Is heterotetramerization also applicable to Commd10- Commd2 complex?. Is tetramerization a property of heterodimers?.

2) COMMD-CCC interactions:

COMMD domains of Commd1, Commd7 and Commd10 interact directly with the NN-CH domain of CCD22 and CCDC93 whereas Commd9 binds specifically to CCDC93 (Figure 8). The crystal structure of COMMD9 presented in this work and the homology model of COMMD1 (Figure 9B) should facilitate the identification of a similar surface patch, distinct from the conserved dimerization core, responsible for binding to the NN-CH domain of CCDC93. Authors could exploit this structural information together with site-directed mutagenesis and bio-layer interferometry as shown in Figure 8C, D to identify and confirm the binding interface. This would add a great value to the present work.

3) Membrane association:

The basic patch (K129, R133, H134, K167) identified in COMM1 that is important for interaction with model membranes, is not conserved in Commd7 or Commd10, which also interact with membranes (Figure 9A). Furthermore, this patch is neither conserved in other Commd members (Figure 3B) suggesting a singular and partial mechanism only applicable to Comm1. How other Commd complexes, dimers or heterodimers, are recruited to the membrane remain unknown. One possibility for membrane recruitment might be through interaction with the CCC complex but exploring this possibility would require the identification of the conserved patch in COMMD proteins for binding to CCDC22 and/or CCDC93.

4) Figure 5: The graphs on top are too small to recognize anything. More importantly, the scattering curves and the resulting envelopes look very similar. Given the ambiguities in fitting such low resolution envelopes, the authors should not make any firm conclusions on the details of the assemblies, since they are most likely wrong. The firm conclusions of these measurements appear to be: Homomeric COMMDs form a dimer in SAXS. In some likelihood, the COMMD domains are in the center whereas the HN domains are at the periphery.. the authors are recommended to tone down the conclusions of different assemblies and maybe only show data and the model of Commd9 in the main manuscript.

5) Figure 7A: Commd8: There is hardly any protein of the correct size in the Commassie gel, and it appears even unclear whether the proposed band is at all related to COMM8. Strangely enough, this protein still interacts with high efficiency with all Commd proteins. That does not make much sense: Either improve or remove.

6) Figure 8A: Again, hardly any CCDC22 full length is expressed and a strong interaction with COMM1 is observed (both with the N- and C-terminal domains of CCDC22 – the authors should also comment on this in the main manuscript). Furthermore, COMM1 also appears to be partly pulled down by GST. This is worrisome, there seems to be an unspecific binding of COMM1 to almost anything and the data for CCDC22 are uninterpretable. These experiments should be improved, for example with more stringent washing conditions or finding better expression conditions.

7) The structural homology analysis of COMMD proteins to Pur-α, CT584 and Cpn0803, do not contribute much to the overall story and are distractive. In this sense, the homology only hints at a shared fold for dimerization but there is no support for functional similarities to Pur-α, or mimicking function by the chlamydial proteins. This section might be shortened and/or incorporated into supplementary material.

8) The liposome binding experiments appear to lack a negative control – i.e. liposomes that are not acidic such as those of pure PC.

[Editors' note: further revisions were requested prior to acceptance, as described below.]

Thank you for resubmitting your work entitled "Structural insights into the architecture and membrane interactions of the conserved COMMD proteins" for further consideration at *eLife*. Your revised article has been favorably evaluated by a Senior Editor and a Reviewing Editor.

The manuscript has been improved but there are some remaining issues about the text that the reviewers agree need to be addressed before acceptance, as outlined below:

1) In the summary the authors make a claim on defining the core principles of Commd homo- and heteromeric assembly. This is clearly an overstated claim as the interaction between Commd proteins seems to be quite promiscuous and still there are no clear principles of how homodimer-versus-heterodimer preference is achieved.

2) Crosslinking data: The WRVD motif should be better introduced. The cross-linked results are interesting and narrow down the potential interface between Commd9 and CCDC93, yet the proposed WRVD motif seems to be part of the conserved dimerisation core (Figure 2). Based on the calponin homology domain of CCDC93, the structure of Commd9, the BS3 cross linking distance, and conserved residues on the surface, authors should be able to propose a plausible contact area in Commd9 despite the NMR unfruitful results.

3) Figure 4: I am still not happy to see three different molecular models for full length Commd1, Commd7 and Commd9. It's just misleading, the envelopes look all the same. One representative model showing that the COMMD and HN domain fit into all envelopes would be completely sufficient and less confusing.

4) Following our initial advice, Figure 5 should go to the supplement.

5) Typo: subsection “COMM domain binds to CCDC22 and CCDC93 through a conserved motif”, first paragraph – should be K133, not K131 (K131 is the residue in CCDC93).

6) Subsection “COMM domain binds to CCDC22 and CCDC93 through a conserved motif” – is missing a full stop.

7) Figure 3—figure supplement 1: aa numbers 151 to 200 are missing.

8) Figure 7 legend: Description for panel D is missing (CCDC93 interaction).

I hope that addressing these points will not be difficult for you. If you can provide a point-by-point summary of the changes you have made I should be able to make a final decision without having to consult the reviewers again.

---

## [Author Response]

Essential revisions:1) COMMD homo- and heterodimers:Homo- and heterodimerization seems to be promiscuous among Commd proteins. While Commd1, Commd7 and Commd9 behave as 1/1 homodimers (Figure 1A), the Commd5-Commd10 complex behaves as a 2/2 heterotetramer (Figure 7E). From Figure 7A, it seems that by co-expression in E. coli, Commd10 binds strongly to Commd2. Is heterotetramerization also applicable to Commd10- Commd2 complex?. Is tetramerization a property of heterodimers?.

The reviewers correctly observed that among the promiscuous Commd-Commd interactions we detect, Commd10 appears to bind strongly to Commd2. We have been able to reconstitute this complex (see Author response image 1), however due to poor yield we have not been able to conduct MALS experiments to determine the precise oligomeric state. Interestingly, this heterodimer does not appear to be a 1:1 complex on gels, although we can’t yet be sure until we improve the purification protocols. We therefore believe that Commd10-Commd2 can behave in a similar manner to the Commd10-Commd5 complex, but we can't yet state if tetramerisation is a common property.

**Author response image 1. respfig1:** Commd2-Commd10 co-purification. GST-Commd2 was co-expressed with Commd9-His, proteins were then purified by GST-affinity chromatography, thrombin cleavage and then His-tag affinity chromatography. The two proteins co-purify in a stable heterodimer (Commd10-His is the upper band and cleaved Commd2 is the lower band.

2) COMMD-CCC interactions:COMMD domains of Commd1, Commd7 and Commd10 interact directly with the NN-CH domain of CCD22 and CCDC93 whereas Commd9 binds specifically to CCDC93 (Figure 8). The crystal structure of COMMD9 presented in this work and the homology model of COMMD1 (Figure 9B) should facilitate the identification of a similar surface patch, distinct from the conserved dimerization core, responsible for binding to the NN-CH domain of CCDC93. Authors could exploit this structural information together with site-directed mutagenesis and bio-layer interferometry as shown in Figure 8C, D to identify and confirm the binding interface. This would add a great value to the present work.

This is an excellent suggestion and we have attempted to define the interface on Commd9 that is coordinating the interaction with NN-CH domain of CCDC93 using cross-linking mass spectrometry. This new data together with text has been included in the revised manuscript. While this data now identifies a region on the Commd9 COMM domain that is involved in binding to the CCDC93 CC-NH, we have not been able to discover a point mutant that blocks this interaction based on this information. Until we are able to determine a high-resolution structure of the complex between the two proteins we would argue that this is likely just because XL-MS is unable to define the process side-chains required for binding.

We also attempted to use NMR to test whether it is possible to assign the backbone of the Commd9 COMM domain for titration mapping experiments with NN-CH domains of CCDC22 and CCDC93. However, the NMR spectra in the conditions tested did not give ^15^N HSQC spectra that can be used for such experiments.

These data are included in the new Figure 8, the previous Figure 8 is now Figure 7.

3) Membrane association:The basic patch (K129, R133, H134, K167) identified in COMM1 that is important for interaction with model membranes, is not conserved in Commd7 or Commd10, which also interact with membranes (Figure 9A). Furthermore, this patch is neither conserved in other Commd members (Figure 3B) suggesting a singular and partial mechanism only applicable to Comm1. How other Commd complexes, dimers or heterodimers, are recruited to the membrane remain unknown. One possibility for membrane recruitment might be through interaction with the CCC complex but exploring this possibility would require the identification of the conserved patch in COMMD proteins for binding to CCDC22 and/or CCDC93.

We agree that it is currently not clear how various members of the COMMD family tether to the membranes. Indeed as our data for Commd1 shows, although it can bind membranes this doesn’t appear to be essential for specific endosomal localization. We actually debated showing this negative data, but eventually felt it was important to show this for completeness. Although speculation, we think that the avidity effect provided by all of the COMMD members and perhaps through direct interaction of other CCC components with membranes and membrane-associated proteins will prove to be key determinants for endosomal recruitment. As discussed above although we now have used XL-MS to gain an insight into the interaction region of Commd9 with CCDC proteins, we are not yet able to generate specific mutants that perturb these interactions to determine the importance in membrane recruitment.

4) Figure 5: The graphs on top are too small to recognize anything. More importantly, the scattering curves and the resulting envelopes look very similar. Given the ambiguities in fitting such low resolution envelopes, the authors should not make any firm conclusions on the details of the assemblies, since they are most likely wrong. The firm conclusions of these measurements appear to be: Homomeric COMMDs form a dimer in SAXS. In some likelihood, the COMMD domains are in the center whereas the HN domains are at the periphery.. the authors are recommended to tone down the conclusions of different assemblies and maybe only show data and the model of Commd9 in the main manuscript.

This figure has been made larger to more clearly show the data. We have also changed the language in the text to suggest that SAXS data confirms the MALLS and crystallography data that COMMD proteins are homodimers in solution. We agree that bold interpretations cannot be made from the scattering experiments, however at the least the data clearly shows the overall shape of various COMMD proteins, and confirms their homodimeric nature. Therefore, we have for now opted to keep all of the SAXS data in this revised version.

The old Figure 5 is now replaced with Figure 4.

5) Figure 7A: Commd8: There is hardly any protein of the correct size in the Commassie gel, and it appears even unclear whether the proposed band is at all related to COMM8. Strangely enough, this protein still interacts with high efficiency with all Commd proteins. That does not make much sense: Either improve or remove.

GST-Commd8 has a tendency to degrade over time, but we have identified through time course experiments that the marked band is indeed GST fused Commd8. The protein appears to be binding other Commd proteins and fragments, as seen in the western blotting-based detection method we have employed that is quite sensitive. However, on the advice of reviewers we have removed this data for Commd8.

Figure 7 has been changed to Figure 6 due to merging of Figure 1 and 2.

6) Figure 8A: Again, hardly any CCDC22 full length is expressed and a strong interaction with COMM1 is observed (both with the N- and C-terminal domains of CCDC22 – the authors should also comment on this in the main manuscript). Furthermore, COMM1 also appears to be partly pulled down by GST. This is worrisome, there seems to be an unspecific binding of COMM1 to almost anything and the data for CCDC22 are uninterpretable. These experiments should be improved, for example with more stringent washing conditions or finding better expression conditions.

New text describing the interactions of COMM domains with both HN and COMM domain has been included in the manuscript. With regards to the full length CCDC22 protein band, like Commd8, CCDC22 also has a tendency to degrade in to smaller fragments. However, there is still some fraction of intact GST fused CCDC22 visible on the Coomassie stained gel as shown by the asterisk and since the detection method is very sensitive, we can still detect an interaction band.

We have also repeated the pull down again and have included the new blot in the revised manuscript. In this new round, the non-specific GST binding has been reduced and comparison with other bands on the gel clearly shows that the interaction to CCDC proteins is specific. We also note that GST is expressed heavily compared to the other bait proteins.

Please note, Figure 8 is now Figure 7.

7) The structural homology analysis of COMMD proteins to Pur-α, CT584 and Cpn0803, do not contribute much to the overall story and are distractive. In this sense, the homology only hints at a shared fold for dimerization but there is no support for functional similarities to Pur-α, or mimicking function by the chlamydial proteins. This section might be shortened and/or incorporated into supplementary material.

We have truncated this section in the text.

8) The liposome binding experiments appear to lack a negative control – i.e. liposomes that are not acidic such as those of pure PC.

We have now included the data containing PC as a negative control.

[Editors' note: further revisions were requested prior to acceptance, as described below.]

The manuscript has been improved but there are some remaining issues about the text that the reviewers agree need to be addressed before acceptance, as outlined below:1) In the summary the authors make a claim on defining the core principles of Commd homo- and heteromeric assembly. This is clearly an overstated claim as the interaction between Commd proteins seems to be quite promiscuous and still there are no clear principles of how homodimer-versus-heterodimer preference is achieved.

We have toned down this statement in the summary to “Here, we have systematically characterised the interactions between several human COMMD proteins, and determined structures of COMMD proteins using X-ray crystallography and X-ray scattering to provide insights into the underlying mechanisms of homo- and heteromeric assembly.” We agree that our data still cannot provide a concise explanation for why specific homo and heterodimers are formed, but it does reveal new insights into how these interactions occur.

2) Crosslinking data: The WRVD motif should be better introduced. The cross-linked results are interesting and narrow down the potential interface between Commd9 and CCDC93, yet the proposed WRVD motif seems to be part of the conserved dimerisation core (Figure 2). Based on the calponin homology domain of CCDC93, the structure of Commd9, the BS3 cross linking distance, and conserved residues on the surface, authors should be able to propose a plausible contact area in Commd9 despite the NMR unfruitful results.

We apologise that this was not described clearly enough. We have now modified the text and also Figure 7 showing the sites of lysine cross-linking to better explain our finding. What we observe is that the two major sites of lysine cross-linking in the Commd9 COMM domain are part of a contiguous surface, and that this surface also includes the residues from the conserved sequence ^128^WRVD^131^. The strictly conserved Trp128 in particular does make key contacts that appear to be required for COMM domain dimerization. But, unusually for a Trp side-chain, it is also surface exposed, suggesting it may be part of a binding surface for the CCDC93 NN-CH domain. However, we did not want to speculate too much based on our data and have therefore modified the text as follows:

“Notably, K133 and K152 are located on contiguous surfaces of the β-sheets of the COMM domains (Figure 7D), suggesting a likely binding surface for the CCDC93 NN-CH domain. This surface also includes the side-chains of the conserved residues ^128^WRVD^131^, with Trp128 in particular being strictly conserved across the entire Commd family (Figure 2B).”

3) Figure 4: I am still not happy to see three different molecular models for full length Commd1, Commd7 and Commd9. It's just misleading, the envelopes look all the same. One representative model showing that the COMMD and HN domain fit into all envelopes would be completely sufficient and less confusing.

We have removed the data showing the SAXS-derived solution structures of Commd1 and Commd7 into Figure 4—figure supplement 2, and retained only the SAXS-derived solution structure of Commd9 in the main text for clarity. We hope this satisfies the reviewer’s concerns.

4) Following our initial advice, Figure 5 should go to the supplement.

Figure 5 has now been moved to Figure 4—figure supplement 4. Subsequent figures have been renumbered accordingly, and figure citations in the text have been modified to reflect the changes.

5) Typo: subsection “COMM domain binds to CCDC22 and CCDC93 through a conserved motif”, first paragraph – should be K133, not K131 (K131 is the residue in CCDC93).

This has been corrected.

6) Subsection “COMM domain binds to CCDC22 and CCDC93 through a conserved motif” – is missing a full stop.

This has been corrected.

7) Figure 3—figure supplement 1: aa numbers 151 to 200 are missing.

This has been corrected.

8) Figure 7 legend: Description for panel D is missing (CCDC93 interaction).

This has been corrected.